# RETHINKING THE GENERALIZATION OF DRUG TARGET AFFINITY PREDICTION ALGORITHMS VIA SIMILARITY AWARE EVALUATION

**Chenbin Zhang**[*1]**, Zhiqiang Hu**[*✉2]**, Chuchu Jiang**[*3]**, Wen Chen**[4]**, Jie Xu**[3]**, Shaoting Zhang**[3]

[1]MoleculeMind, [2]Peking University, [3]Shanghai AI Laboratory, [4]SenseTime Research

```
chenbinzhang@moleculemind.com,huzq@pku.edu.cn,
{jiangchuchu,xujie,zhangshaoting}@pjlab.org.cn,
chenwen@sensetime.com
```

## ABSTRACT

Drug-target binding affinity prediction is a fundamental task for drug discovery. It has been extensively explored in literature and promising results are reported. However, in this paper, we demonstrate that the results may be misleading and cannot be well generalized to real practice. The core observation is that the canonical randomized split of a test set in conventional evaluation leaves the test set dominated by samples with high similarity to the training set. The performance of models is severely degraded on samples with lower similarity to the training set but the drawback is highly overlooked in current evaluation. As a result, the performance can hardly be trusted when the model meets low-similarity samples in real practice. To address this problem, we propose a framework of similarity aware evaluation in which a novel split methodology is proposed to adapt to any desired distribution. This is achieved by a formulation of optimization problems which are approximately and efficiently solved by gradient descent. We perform extensive experiments across five representative methods in four datasets for two typical target evaluations and compare them with various counterpart methods. Results demonstrate that the proposed split methodology can significantly better fit desired distributions and guide the development of models. Code is released at https://github.com/Amshoreline/SAE/tree/main.

## 1 INTRODUCTION

Drug-target binding affinity (DTA) prediction is a fundamental and crucial task for drug discovery. It evaluates the effectiveness of drug candidates, or samples, and sees its application in a large-scale virtual screening where most ineffective candidates are filtered out to save experimental cost and time (Chatterjee et al., 2023). DTA is quantitatively measured by inhibition constant Ki, half maximal inhibitory concentration IC50, etc., which are all real-valued (Monteiro et al., 2022). The prediction performance is commonly evaluated by mean absolute error (MAE) and coefficient of determination ($R^2$).

The task of DTA prediction has been extensively studied for decades (Chen et al., 2018; Askr et al., 2023). Related works can be categorized as structure-based, sequence-based, and similarity-based (Wu et al., 2018; Chuang et al., 2020). Structure-based methods rely on 3D structures of samples, target proteins, or their complexes. Although theoretically accessible to most comprehensive information following the dogma "structure determines function", structure-based methods are limited by available 3D structures, especially experimentally verified structures, and also hindered in practice by poor time efficiency. In contrast, sequence-based and similarity-based methods are fast and do not set 3D structures as prerequisite (Xu et al., 2017; Zhang et al., 2022). Instead, they take

---

[*]Equal contribution.
[✉]Corresponding Author.

as input residual sequences, Simplified Molecular-Input Line-Entry System (SMILES) sequences, fingerprint sequences, atom-bond graphs, or the derived pairwise similarities, which are easier to acquire with lower cost. Moreover, these sequences and similarities are readily processed by diversified sophisticated backbones including convolutional neural networks (CNNs) (Öztürk et al., 2018; Li et al., 2019; Hu et al., 2023), recurrent neural networks (RNNs) (Karimi et al., 2019; Yuan et al., 2022), graph neural networks (GNNs) (Nguyen et al., 2021; Yang et al., 2022; Tang et al., 2022; Wang et al., 2022) and transformers (Chithrananda et al., 2020; Zhao et al., 2022; Song et al., 2023; Jiang et al., 2023), and enjoy the benefits of the development of deep learning techniques. As a result, sequence-based and similarity-based methods are shown to reach new high performance and are drawing increasing attention.

Although promising results are reported, we find, surprisingly, that these results may be misleading. Take the task of IC50 prediction for target EGFR as an example, as shown in Figure 1, we evaluate five state-of-the-art and representative methods and the best-performing one, SAM-DTA (Hu et al., 2023), achieves a MAE of 0.6012 and $R^2$ of 0.6505 for the *whole* test set. However, if we dive into the performance and divide the test set according to the similarity of the sample to the training set, we find a clear performance degradation for low-similarity samples: the MAE deteriorates to 1.2970 for samples with similarity less than 1/3 and $R^2$ to -0.6385. The gap is huge. Nevertheless, poor performance on low-similarity samples does not affect the *whole* performance since they only occupy a negligible proportion: only 16 samples with similarity less than 1/3 out of a total of 873 samples in the test set (Figure 1). In other words, the test set is dominated by high-similarity samples and performance for low-similarity samples is overwhelmed in current evaluations. We will show that the phenomenon exists across different similarity measures, performance metrics, datasets, and methods, and therefore it is general. Consequently, the evaluation will be misleading to practitioners, especially when the trained model meets low-similarity samples when used in real practice.

We argue that the core of the problem lies in the canonical randomized split of the test set. Randomized split follows the assumption of independent identical distribution (I.I.D.), which is the foundation of most statistical learning theories. However, in drug discovery samples are not necessarily independent of each other: in practice, mutually similar variants are more likely to be tested together in high-throughput experiments, while at the same time, they have to avoid high similarity to approved drugs for intellectual property issues (Harren et al., 2024). Empirically, drugs developed at different times show significant distinction in their properties (Sheridan et al., 2022). As a result, practitioners would not always expect the samples they are going to test to follow the same distribution as historical samples. This in turn raises a request to model development that the test set should satisfy a desired distribution. For example, one may need a test set that is uniform at different similarity bins; others may ask the test set samples to be all limited within predefined similarity bounds, and so on (Li & Yang, 2017; Simm et al., 2021; Luo et al., 2024; Tricarico et al., 2024).

We formulate the test set split with a desired distribution as a combination optimization problem. The problem is infeasible to solve for optimum due to efficiency issues. In this work, we address this challenge by relaxing it to a continuous optimization problem where samples are allowed to coexist in the training set and test set with a "probability" or weight. Further, the objective function contains non-differentiable operations including taking the maximum and counting in similarity bins, and are approximated in this work by differentiable counterparts. We will show that the degree of approximation is adjustable by introduced hyper-parameters. Next, the resulting optimization problem has no closed-form solution, and hence we have resorted to Lagrangian multipliers with a numerical method implemented by PyTorch and Cooper (Gallego-Posada & Ramirez, 2022). Finally, we analyze the continuously-valued solution and find the non-negligible approximation error induced by the relaxation. To this end, we introduce a regularization term that penalizes samples whose weight is far from bipartition. We refer to our strategy as **S**imilarity **A**ware **E**valuation, abbreviated as SAE. By doing all this, we have managed to achieve test set splits with various desired distributions.

Extensive experiments are performed to substantiate the effectiveness of our split strategy. To begin with, our split strategy can achieve a uniformly distributed test set across various similarity bins (Figure 1). Subsequently, we evaluate the performance of five DTA prediction methods on this test set. The results underscore a distinct relationship between the performance and the corresponding similarity levels, suggesting a more comprehensive assessment of balanced split across varied methods in comparison to the strategy of randomized split. Moreover, in scenarios where the samples practitioners intend to test deviate from the distribution of existing samples, our split strategy can effectively split the training set and internal test set based on the similarity distribution of the test

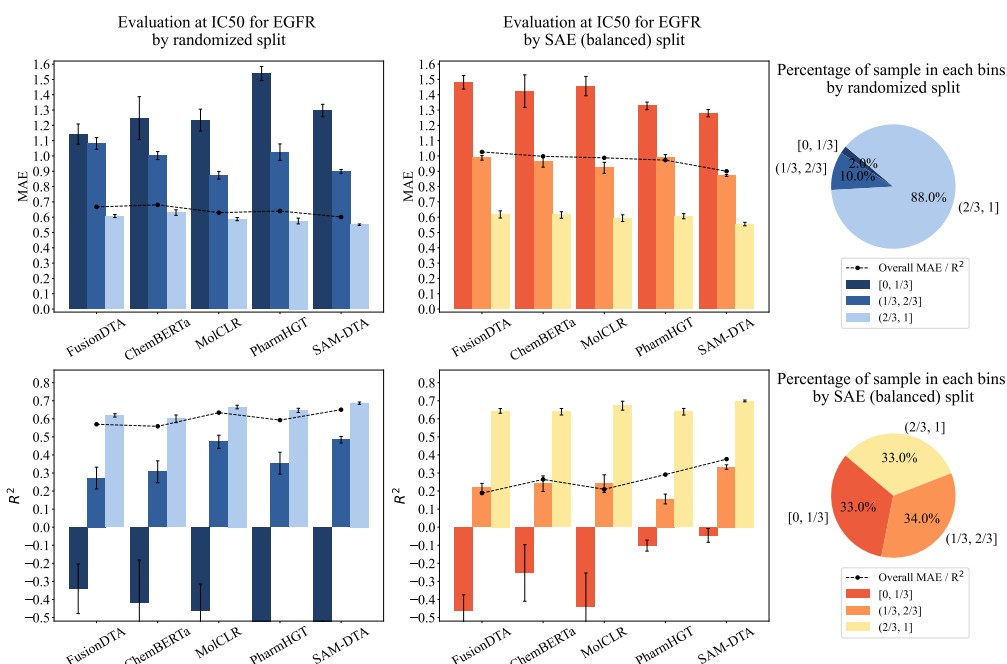

Figure 1: Comparison of randomized split and SAE (balanced) split at IC50 for EGFR. The randomized split led to 88% of test samples yielding a high similarity ($> 2/3$) to the training set. In contrast, our SAE (balanced) split strategy ensures a more balanced distribution of similarities. The evaluation of five DTA prediction methods demonstrates that the performance aligns with the similarity levels. In the randomized split, the overall performance closely resembles that of high-similarity samples, thus failing to evaluate the performance when encountering low-similarity samples.

set (Figure 4). Here we conduct hyper-parameter searches on the internal test set across five DTA prediction methods to assess the efficacy of our split strategy. Compared to previous split strategies, our split strategy facilitates the selection of optimal hyper-parameters, enhancing performance on the external test set (Figure 5). In other scenarios where practitioners specify predefined similarity constraints for the test set samples, such as a maximum similarity limit of 0.4 or 0.6, or even a range bounded by a minimum and maximum similarity of 0.4 and 0.6, our split strategy ensures the majority of samples in the test set adhering to these requirements (Figure 6).

## 2    PROBLEM OF RANDOMIZED SPLIT

In this section, we will show that the imbalanced distribution of samples and the consequent overshadowing of low-similarity samples is general for a randomized split of a test set across different similarity measures, performance metrics, datasets, and methods. We will first give the details of the example in Figure 1 and then explore possible variants.

In the example demonstrated by Figure 1, we take the IC50 dataset of target EGFR with a total of 4,361 samples, which is one of the largest datasets we are able to find. The dataset has originated from the BindingDB database, and IC50 has been converted to its negative logarithm form, $pIC50 = -\log_{10} IC50\ (Molar)$ following the convention. Then, we randomly split the dataset into a training set and test set with a ratio of 8:2. Subsequently, we train and validate the DTA prediction models on the training set and evaluate their performance on the test set. It should be noted that we follow the original hyper-parameter tuning procedure in the source code of each DTA prediction method.

To perform the fine-grained evaluation with respect to the similarity, we first define the pairwise similarity for sample $x_1$ and $x_2$,

$$PairwiseSimilarity(x_1, x_2) = SimilarityMeasure\left(Feature(x_1), Feature(x_2)\right) \qquad (1)$$

and then derive the similarity to the union of the training set by aggregation, for sample $x \in TestSet$.

Table 1: Variations of *SimilarityToTrainingSet* related to feature extraction, similarity measure, aggregation functions, and performance metrics. We choose PharmHGT and SAM-DTA as the example methods for the detailed showcase.

| | Randomized Split (MAE) | | | | | |
|---|---|---|---|---|---|---|
| Bin | Feature: RDKit fingerprint | | | Feature: Avalon fingerprint | | |
| | Count (Ratio) | PharmHGT | SAM-DTA | Count (Ratio) | PharmHGT | SAM-DTA |
| [0   , 1/3] | 8 (0.0092) | 1.7551 | 1.6787 | 0 (0.0000) | - | - |
| (1/3, 2/3] | 34 (0.0389) | 1.3214 | 1.0040 | 28 (0.0321) | 1.4646 | 1.3319 |
| (2/3,   1] | 831 (0.9519) | 0.6015 | 0.5743 | 845 (0.9679) | 0.6128 | 0.5770 |
| overall | 873 (1.0000) | 0.6401 | 0.6012 | 873 (1.0000) | 0.6401 | 0.6012 |
| | SimilarityMeasure: Sokal similarity | | | SimilarityMeasure: Dice coefficient | | |
| [0   , 1/3] | 33 (0.0378) | 1.5051 | 1.2444 | 0 (0.0000) | - | - |
| (1/3, 2/3] | 398 (0.4559) | 0.7066 | 0.6619 | 33 (0.0378) | 1.5051 | 1.2444 |
| (2/3,   1] | 442 (0.5063) | 0.5157 | 0.4985 | 840 (0.9622) | 0.6061 | 0.5759 |
| overall | 873 (1.0000) | 0.6401 | 0.6012 | 873 (1.0000) | 0.6401 | 0.6012 |
| | Aggregation: Top-3 | | | Aggregation: Top-5 | | |
| [0   , 1/3] | 17 (0.0195) | 1.5188 | 1.3149 | 24 (0.0275) | 1.5627 | 1.3469 |
| (1/3, 2/3] | 171 (0.1959) | 0.8890 | 0.7748 | 240 (0.2749) | 0.8014 | 0.7228 |
| (2/3,   1] | 685 (0.7847) | 0.5562 | 0.5401 | 609 (0.6976) | 0.5402 | 0.5239 |
| overall | 873 (1.0000) | 0.6401 | 0.6012 | 873 (1.0000) | 0.6401 | 0.6012 |
| | Randomized Split ($R^2$) | | | | | |
| Bin | Feature: RDKit fingerprint | | | Feature: Avalon fingerprint | | |
| | Count (Ratio) | PharmHGT | SAM-DTA | Count (Ratio) | PharmHGT | SAM-DTA |
| [0   , 1/3] | 8 (0.0092) | 0.1555 | 0.2579 | 0 (0.0000) | - | - |
| (1/3, 2/3] | 34 (0.0389) | -0.0371 | 0.3529 | 28 (0.0321) | 0.1028 | 0.2921 |
| (2/3,   1] | 831 (0.9519) | 0.6327 | 0.6706 | 845 (0.9679) | 0.6169 | 0.6672 |
| overall | 873 (1.0000) | 0.5928 | 0.6505 | 873 (1.0000) | 0.5928 | 0.6505 |
| | SimilarityMeasure: Sokal similarity | | | SimilarityMeasure: Dice coefficient | | |
| [0   , 1/3] | 33 (0.0378) | -0.1562 | 0.1866 | 0 (0.0000) | - | - |
| (1/3, 2/3] | 398 (0.4559) | 0.5412 | 0.6057 | 33 (0.0378) | -0.1562 | 0.1866 |
| (2/3,   1] | 442 (0.5063) | 0.6942 | 0.7156 | 840 (0.9622) | 0.6290 | 0.6711 |
| overall | 873 (1.0000) | 0.5928 | 0.6505 | 873 (1.0000) | 0.5928 | 0.6505 |
| | Aggregation: Top-3 | | | Aggregation: Top-5 | | |
| [0   , 1/3] | 17 (0.0195) | -0.6280 | -0.3148 | 24 (0.0275) | -0.2379 | 0.0891 |
| (1/3, 2/3] | 171 (0.1959) | 0.4283 | 0.5743 | 240 (0.2749) | 0.4756 | 0.5794 |
| (2/3,   1] | 685 (0.7847) | 0.6483 | 0.6712 | 609 (0.6976) | 0.6584 | 0.6804 |
| overall | 873 (1.0000) | 0.5928 | 0.6505 | 873 (1.0000) | 0.5928 | 0.6505 |

$$SimlarityToTrainingSet(x) = \underset{t \in TrainingSet}{Aggregation} \ PairwiseSimlarity(x, t) \qquad (2)$$

where in the example of Figure 1, we set *Feature* as the Morgan fingerprint and *SimilarityMeasure* as the Tanimoto coefficient, which are both commonly used to measure the similarity of samples, and we set *Aggregation* as the maximum function (Bajusz et al., 2015; Ying et al., 2021).

Next, we compare other variants for these functions. For the feature extractor *Feature*, we compare other widely used molecular descriptors including Avalon fingerprint and RDKit fingerprint (a.k.a. topological fingerprint); for the function *SimilarityMeasure* we compare Sokal similarity and Dice coefficient, which are both symmetric for its parameters; and finally for the *Aggregation* function, we compare the general top-$k$ averaging where the maximum function can be seen as a special case

Table 2: Comparison of randomized split and SAE (balanced) split at IC50 for BACE1, Ki for Carbonic anhydrase I and Carbonic anhydrase II. We choose PharmHGT and SAM-DTA as the example methods for the detailed showcase.

| | IC50 for Target BACE1 (MAE) | | | | | |
|---|---|---|---|---|---|---|
| Bin | Randomized Split | | | SAE (balanced) Split | | |
| | Count (Ratio) | PharmHGT | SAM-DTA | Count (Ratio) | PharmHGT | SAM-DTA |
| [0 , 1/3] | 10 (0.0108) | 1.3743 | 1.1204 | 309 (0.3330) | 1.1397 | 1.0309 |
| (1/3, 2/3] | 67 (0.0722) | 0.6334 | 0.6928 | 311 (0.3351) | 0.6410 | 0.6693 |
| (2/3, 1] | 851 (0.9170) | 0.4611 | 0.4594 | 308 (0.3319) | 0.4747 | 0.4808 |
| overall | 928 (1.0000) | 0.4834 | 0.4834 | 928 (1.0000) | 0.7518 | 0.7272 |

| | IC50 for Target BACE1 ($R^2$) | | | | | |
|---|---|---|---|---|---|---|
| [0 , 1/3] | 10 (0.0108) | -0.0553 | 0.3702 | 309 (0.3330) | -0.2983 | -0.1261 |
| (1/3, 2/3] | 67 (0.0722) | 0.6789 | 0.6439 | 311 (0.3351) | 0.5848 | 0.5637 |
| (2/3, 1] | 851 (0.9170) | 0.7113 | 0.7150 | 308 (0.3319) | 0.7797 | 0.7803 |
| overall | 928 (1.0000) | 0.7190 | 0.7256 | 928 (1.0000) | 0.5329 | 0.5665 |

| | Ki for Target Carbonic anhydrase I (MAE) | | | | | |
|---|---|---|---|---|---|---|
| Bin | Randomized Split | | | SAE (balanced) Split | | |
| | Count (Ratio) | PharmHGT | SAM-DTA | Count (Ratio) | PharmHGT | SAM-DTA |
| [0 , 1/3] | 7 (0.0079) | 1.1467 | 0.8798 | 264 (0.2983) | 0.8410 | 0.8729 |
| (1/3, 2/3] | 205 (0.2316) | 0.5843 | 0.6605 | 311 (0.3514) | 0.6706 | 0.6877 |
| (2/3, 1] | 673 (0.7605) | 0.4986 | 0.4896 | 310 (0.3503) | 0.6039 | 0.5740 |
| overall | 885 (1.0000) | 0.5236 | 0.5323 | 885 (1.0000) | 0.6981 | 0.7031 |

| | Ki for Target Carbonic anhydrase I ($R^2$) | | | | | |
|---|---|---|---|---|---|---|
| [0 , 1/3] | 7 (0.0079) | -0.3232 | 0.1308 | 264 (0.2983) | -0.0389 | -0.0282 |
| (1/3, 2/3] | 205 (0.2316) | 0.5733 | 0.4820 | 311 (0.3514) | 0.3642 | 0.3478 |
| (2/3, 1] | 673 (0.7605) | 0.5037 | 0.5270 | 310 (0.3503) | 0.3917 | 0.4262 |
| overall | 885 (1.0000) | 0.5257 | 0.5174 | 885 (1.0000) | 0.2994 | 0.3071 |

| | Ki for Target Carbonic anhydrase II (MAE) | | | | | |
|---|---|---|---|---|---|---|
| Bin | Randomized Split | | | SAE (balanced) Split | | |
| | Count (Ratio) | PharmHGT | SAM-DTA | Count (Ratio) | PharmHGT | SAM-DTA |
| [0 , 1/3] | 8 (0.0087) | 0.5879 | 0.6645 | 244 (0.2667) | 1.0450 | 1.0564 |
| (1/3, 2/3] | 201 (0.2197) | 0.6807 | 0.7009 | 342 (0.3738) | 0.7389 | 0.7572 |
| (2/3, 1] | 706 (0.7716) | 0.5615 | 0.5426 | 329 (0.3596) | 0.6172 | 0.5813 |
| overall | 915 (1.0000) | 0.5879 | 0.5785 | 915 (1.0000) | 0.7768 | 0.7738 |

| | Ki for Target Carbonic anhydrase II ($R^2$) | | | | | |
|---|---|---|---|---|---|---|
| [0 , 1/3] | 8 (0.0087) | 0.6739 | 0.4277 | 244 (0.2667) | -0.0885 | 0.0087 |
| (1/3, 2/3] | 201 (0.2197) | 0.5803 | 0.5742 | 342 (0.3738) | 0.4657 | 0.4690 |
| (2/3, 1] | 706 (0.7716) | 0.5509 | 0.5932 | 329 (0.3596) | 0.4760 | 0.5346 |
| overall | 915 (1.0000) | 0.5684 | 0.5938 | 915 (1.0000) | 0.3776 | 0.4192 |

of $k = 1$. Note that averaging or taking the median over the whole training set is not suitable. This is because the majority of samples in the training set have a low similarity to a specific sample, and averaging or taking the median over the whole training set is not able to tell whether there exists any high-similarity ones. The results are shown in Table 1. Here we choose two example methods for detailed showcase (PharmHGT (Jiang et al., 2023) and SAM-DTA (Hu et al., 2023)), while the results of other methods can be found in the appendix.

For the prediction method, as shown in Figure 1, we select five state-of-the-art and representative DTA prediction methods. Molecular Contrastive Learning of Representations (MolCLR) sees sam-

ples as atom-bond graphs and employs GCN and GIN to learn the molecular representations by contrastive pairs (Wang et al., 2022). Sequence-agnostic model for drug-target binding affinity prediction (SAM-DTA), on contrast, takes as input the Simplified Molecular-Input Line-Entry System (SMILES) of samples and processes it using 1D-CNN with dilated parallel residual blocks (Hu et al., 2023). SMILES is also utilized in FusionDTA but is processed by a unified LSTM model with linear attention mechanism (Yuan et al., 2022). Finally, we include two transformer-based methods. One is ChemBERTa which takes as input SMILES of samples and builds a model with 12 attention heads and 6 layers (Chithrananda et al., 2020). The other is PharmHGT which leverages a unique pharmacophoric-constrained heterogeneous molecule graph and two various transformers to extract chemical properties and predict molecular attributes (Jiang et al., 2023).

We also investigate the problem with other tasks and datasets. Specifically, for the task of IC50 prediction, we also perform experiments in the dataset of target BACE1 with a total of 4,636 samples, and we further extend the experiments to the task of Ki prediction for targets Carbonic anhydrase I and Carbonic anhydrase II, with 5,307 and 5,487 samples respectively. For all of these datasets, we apply the same preprocessing as that of target EGFR, except that taking the negative logarithm form is not applicable to Ki datasets. The results are collectively presented in Table 2. Here we choose PharmHGT and SAM-DTA as the example methods for a detailed showcase, while the comprehensive collection of results can be found in the appendix.

In summary, extensive experiments demonstrate the generality of the imbalanced distribution of samples caused by randomized split and the consequent overshadowing of low-similarity samples. This problem will be analyzed and addressed in the following section.

## 3 SIMILARITY AWARE EVALUATION

In this section, we will elaborate on the proposed Similarity Aware Evaluation (SAE) which aims at obtaining a test set with the desired distribution. We will exemplify the method for a test set that is uniform at similarity-based bins (see Figure 1 for 3 similarity-based bins), and then extend it to other desired distributions.

The split for a test set that is uniform across similarity-based bins can be formulated as a combinatorial optimization problem as follows. Given a dataset $X = \{x_i, i = 1, 2, ..., N\}$, a pairwise similarity matrix $\{s_{ij} \in [0, 1], s_{ii} = 0, i = 1, 2, ..., N; j = 1, 2, ..., N\}$, a ratio $\alpha$, and $K$ bins with boundaries $\{b_k, k = 0, 1, 2, ..., K\}$, find a subset (test set) $X_{ts} \subset X, |X_{ts}| = \alpha N$, such that

$$f(X_{ts}) = \sum_{k=1}^{K} \frac{(o_k - \alpha N/K)^2}{\alpha N/K} \tag{3}$$

is minimized, where

$$o_k = |\{x_i \in X_{ts} : b_{k-1} < r_i \le b_k\}| \tag{4}$$

$$r_i = \max_{x_j \in X_{tr}} s_{ij} \tag{5}$$

$$X_{tr} = X - X_{ts} \tag{6}$$

$X_{tr}$ denotes the training set, $r_i$ the similarity of $x_i$ to the training set, and $o_k$ the count for each of the $K$ bins. Note that the objective function $f$ is essentially the $\chi^2$ statistics in the Chi-Square ($\chi^2$) Test, where $o_k$ is the observed count in each bin and $\alpha N/K$ is expected. Note also that we specially set $s_{ii} = 0$ in the pairwise similarity matrix. This has no effect on the problem itself but can avoid that $r_i$ falls trivially to 1 due to the maximum operation for the relaxed problem below.

The combination optimization problem is infeasible to solve for optimum. As a result, we relax it to a continuous optimization problem where samples are allowed to coexist in the training set and test set by the introduction of the weights $\{w_i \in [0, 1], i = 1, 2, ..., N\}$ and by $|X_{ts}| = \alpha N$ we have constraints $\sum_i w_i = \alpha N$. Next, we have to deal with non-differentiable operations in the objective function $f$ including taking the maximum and counting in similarity-based bins. For the maximum function in the calculation of $r_i$, we approximate it by the LogSumExp operation with a hyper-parameter $\beta$,

$$r_i = \max_{x_j \in X_{tr}} s_{ij} = \max_j (1 - w_j) s_{ij} \approx \frac{1}{\beta} \log \sum_j \exp\left(\beta (1 - w_j) s_{ij}\right) \tag{7}$$

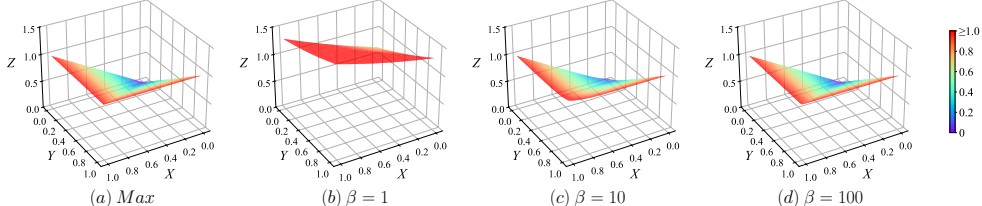

Figure 2: Impact of the hyper-parameter $\beta$ on the approximation of the maximum function in Eq. 7. To illustrate this impact, we consider a simplified scenario involving only two random variables: $X$ and $Y$. (a) $Z = Max(X, Y)$; (b-d) $Z = 1/\beta \log(\exp(\beta X) + \exp(\beta Y))$. A larger value of $\beta$ results in a more accurate approximation, with $\beta = 100$ yielding an excellent result.

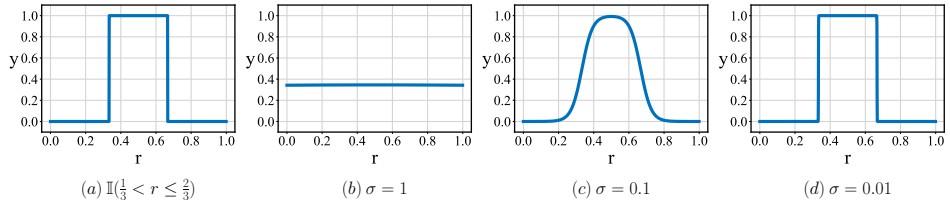

Figure 3: Influence of the hyper-parameter $\sigma$ in Eq. 11. we analyze a specific case where $K = 3, b_k = k/3, c_k = (2k - 1)/6$. (a) $y = \mathbb{I}(b_1 < r \leq b_2)$; (b-d) $y = \exp\left((-(r - c_2)^2/(2\sigma^2))\right) / \sum_{k=1}^{3} \exp\left(-(r - c_k)^2/(2\sigma^2)\right)$. A decrease in the value of $\sigma$ leads to a more precise estimation, with $\sigma = 0.01$ producing an outstanding result.

In terms of counting for similarity-based bins in the calculation of $o_k$, we approximate the discrete event of a sample falling into a specific bin by a continuous score which depends on how far $r_i$ of the sample deviates from the center of the bin. The score function is defined following the bell-shaped normal distribution with the center of the bin as the expectation and a tunable standard deviation. For a sample, scores across all bins are normalized. Specifically, denote $c_k = (b_{k-1} + b_k)/2$ as the center of each bin, $\sigma_k$ as the tunable standard deviation, we have:

$$o_k = |\{x_i \in X_{ts} : b_{k-1} < r_i \leq b_k\}| \tag{8}$$

$$= \sum_i w_i \mathbb{I}(b_{k-1} < r_i \leq b_k) \tag{9}$$

$$\approx \sum_i w_i \frac{\frac{1}{\sqrt{2\pi}\sigma_k} \exp\left(-(r_i - c_k)^2/(2\sigma_k^2)\right)}{\sum_{k'} \frac{1}{\sqrt{2\pi}\sigma_{k'}} \exp\left(-(r_i - c_{k'})^2/(2\sigma_{k'}^2)\right)} \tag{10}$$

where $\mathbb{I}$ is the indicator function. In this paper, we set $\sigma_k = \sigma, k = 1, 2, ..., K$. Thus, we obtain the following expression:

$$o_k \approx \sum_i w_i \frac{\exp\left(-(r_i - c_k)^2/(2\sigma^2)\right)}{\sum_{k'} \exp\left(-(r_i - c_{k'})^2/(2\sigma^2)\right)} = \sum_i w_i \, \underset{k}{softmax} \left(-\frac{(r_i - c_k)^2}{2\sigma^2}\right) \tag{11}$$

Figure 2 and Figure 3 illustrate the error induced by these two differentiable approximations, with respect to hyper-parameter $\beta$ and $\sigma$, respectively. In Figure 2 we compare $\beta$ between values of 1, 10, and 100 and plot the surface for a special case of maximum over two variables. It can be seen that a larger $\beta$ achieves a better approximation, but is also prone to overflow in practice. We use $\beta = 100$ throughout the paper. For Figure 3, on the other hand, we show the comparison of $\sigma$ values between 1, 0.1, and 0.01 for an example case of the indicator of the second bin for a 3-bin setting $b_k = k/3$. The degree of approximation gets better when the value of $\sigma$ decreases, and is pretty well when $\sigma = 0.01$. For the sake of flexibility, we set $\sigma_k = 0.1(b_k - b_{k-1})$ in rest of the paper.

At the moment we seem to be ready to arrive at the approximated optimization function. However, in practice, we find that the approximation error induced by relaxing $w_i$ from $\{0, 1\}$ to $[0, 1]$ is not negligible. In fact, a considerable proportion of $w_i$ solved is neither near 0 nor 1. To address this

issue, we are inspired by the concept of entropy, and propose to add a regularization term,

$$l_{reg} = -\lambda \sum_i \left( w_i \log(w_i) + (1 - w_i) \log(1 - w_i) \right) \tag{12}$$

where $\lambda$ is a hyper-parameter that balances between the objective function and the regularization term. Finally, we have the optimization problem,

$$\underset{w_i}{minimize} \quad \sum_{k=1}^K \frac{(o_k - \alpha N/K)^2}{\alpha N/K} + l_{reg} \tag{13}$$

$$subject\ to \quad \sum_i w_i = \alpha N \tag{14}$$

$$0 \le w_i \le 1, i = 1, 2, ..., N \tag{15}$$

where

$$o_k = \sum_i w_i \underset{k}{softmax} \left( -\frac{(r_i - c_k)^2}{2\sigma^2} \right) \tag{16}$$

$$r_i = \frac{1}{\beta} \log \sum_j \exp \left( \beta(1 - w_j) s_{ij} \right) \tag{17}$$

$$l_{reg} = -\lambda \sum_i \left( w_i \log(w_i) + (1 - w_i) \log(1 - w_i) \right) \tag{18}$$

Note that the optimization problem has no closed-form solution, and hence we have resorted to Lagrangian multipliers with numerical method implemented by PyTorch and Cooper.

For other desired distributions, one can modify the objective function $f$ in a straightforward way while the approximation tricks and regularization term can be retained, and the resulting optimization function can also be solved by Lagrangian multipliers with numerical method. Generally, if the expected count in each bin is $e_k, k = 1, 2, ..., K$, the objective function can be readily modified as

$$\sum_{k=1}^K \frac{(o_k - e_k)^2}{e_k} + l_{reg} \tag{19}$$

## 4 EXPERIMENTS

### 4.1 BALANCED SPLIT

In Section 2, we demonstrated that within the context of the randomized split, suboptimal performance on low-similarity samples does not significantly impact the overall performance, as they only occupy a negligible proportion. To avoid disregarding samples with low similarity, we implemented a "balanced split" using similarity aware split strategy to achieve a uniformly distributed test set across various similarity bins ($[1/3, 2/3], (1/3, 2/3], (2/3, 1]$). Figure 1 shows a comparison of randomized split and SAE (balanced) split at IC50 for EGFR. The randomized split strategy yielded a case in which 88% of test samples have high similarity ($> 2/3$) to the training set, while our split strategy guarantees a more evenly distributed range of similarities. The evaluation at the SAE (balanced) split reveals that the performance of each model aligns with the respective similarity levels. Hence, our SAE (balanced) split provides a more accurate representation of the performance.

Additional results at other tasks and datasets are delineated in Table 2. Given the space constraint, we provide experimental results of two representative DTA prediction methods. Notably, analogous phenomena are observed across the remaining three datasets. The comprehensive collection of results can be found in the appendix.

### 4.2 MIMIC SPLIT

When prior knowledge about the external dataset—such as the distribution of similarity to existing samples—is available, we can construct an internal test set that mirrors this distribution. This approach enables optimal hyperparameter selection for the DTA prediction method, thereby enhancing performance on the external dataset.

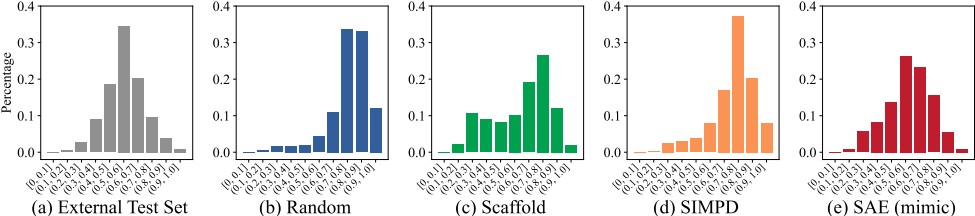

Figure 4: The similarity distribution of the internal test set across different split strategies. (b) Randomized split leads to a scenario where most internal test samples are highly similar to the training set. (c) Scaffold split produces a more balanced distribution. (d) SIMPD split yields a distribution similar to the random split. (e) Our SAE (mimic) split brings the internal test set's distribution closest to that of the external test set.

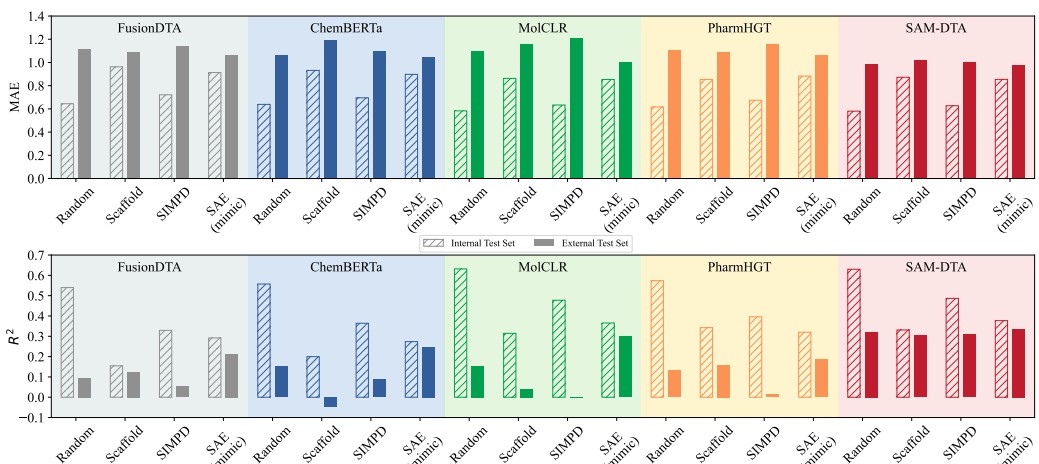

Figure 5: Comparison of the generalization ability of different split strategies at IC50 for EGFR across five DTA prediction methods. The external test set performance of the mimic split surpasses that of other split strategies.

We conducted experiments on the task of predicting IC50 values using the EGFR target dataset. Specifically, 70.2% samples were procured from ChEMBL (Zdrazil et al., 2024), while the remaining samples were obtained from PubChem (Kim et al., 2023), the US Patent and the scientific literature available in BindingDB (Gilson et al., 2016). For our analysis, we classified the ChEMBL-derived samples as internal data, while those obtained from the other sources as the external test set. We first computed the similarity distribution between the external test set and the internal data, as shown in Figure 4 (a). Subsequently, we employed the Randomized split, Scaffold split, and SIMPD split (Landrum et al., 2023) to split the internal data into a training set and an internal test set with a ratio of 70% and 30%. The similarity distributions between the internal test set and the training set for these splits are depicted in Figure 4 (b-d), respectively. Finally, we utilized the proposed strategy to split the internal data, thereby mimicking the similarity distribution observed in the external test set. The results are illustrated in Figure 4 (e). We refer to this split strategy as "mimic split".

We searched for hyper-parameters such as optimizer, learning rate, batch size, and other relevant hyper-parameters. Details of the hyper-parameters for each method are provided in the appendix. Experimental results are shown in Figure 5, our SAE (mimic) split strategy consistently yields optimal hyper-parameter sets for all five DTA prediction methods. Among the various split strategies, the scores of our SAE (mimic) split on the internal test set are the most closely aligned with those on the external test set.

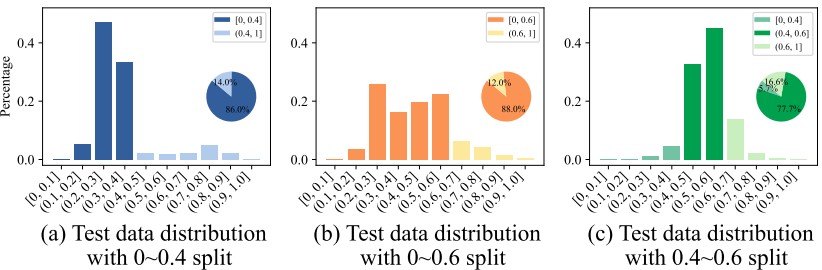

(a) Test data distribution with 0~0.4 split

(b) Test data distribution with 0~0.6 split

(c) Test data distribution with 0.4~0.6 split

Figure 6: Other applications of our split strategy on the IC50 dataset of target EGFR. (a) 86% of the test samples satisfied the desired distribution with a maximum similarity of 0.4, (b) 88% of the test samples met the criteria for a maximum similarity of 0.6, and (c) 77.7% of the test samples fulfilled the requirements for a similarity range between 0.4 and 0.6.

## 4.3 OTHER APPLICATIONS

Beyond achieving balanced splits, our strategy supports distributions with maximum similarities of 0.4 or 0.6 or a range between 0.4 and 0.6. In a 7:3 train-test split on the EGFR target's IC50 dataset (Figure 6), SAE ensured 86%, 88%, and 77.7% of test samples met the criteria, respectively. This underscores the flexibility of the strategy in accommodating diverse split needs. Moreover, since SAE can flexibly achieve desired distributions by capturing the similarity between pairs of data samples, it can also be applied to QSAR scenarios, including ADMET prediction, drug design (De et al., 2022; Tropsha et al., 2024), as well as the prediction of protein-protein interactions (PPI) (Sharma & Bhatia, 2021) and drug-drug interactions (DDI) (Dmitriev et al., 2019).

## 5 RELATED WORKS

When evaluating machine learning methods, it is essential to set aside a test set for benchmarking (Wu et al., 2018). The similarity between the training set and the test set significantly influences the performance of these methods (Sheridan et al., 2004; Cherkasov et al., 2014; Pahikkala et al., 2015; Sieg et al., 2019; Nguyen et al., 2022; Atas Guvenilir & Doğan, 2023; Harren et al., 2024). However, in the field of chemical data, imbalanced data distributions are an inherent and unavoidable challenge (Harren et al., 2024; Yang et al., 2020; Tossou et al., 2024). Therefore, it is crucial to design dataset split strategies that account for these imbalances and ensure meaningful evaluation of model performance (Li & Yang, 2017; Sheridan et al., 2022). The commonly used random splitting method may fail to meet the requirements due to inherent data bias. A typical solution is to exclude all samples in the training set that are similar to those in the test set (Li et al., 2021; Scantlebury et al., 2023; Luo et al., 2024; 2017; Wan et al., 2019; Atas Guvenilir & Doğan, 2023). Recently, several advanced split strategies have been proposed, including scaffold split (Bemis & Murcko, 1996; Fang et al., 2022; Zhou et al., 2023; Liu et al., 2024), time split (Guan et al., 2023; Stärk et al., 2022), stratified split (Wu et al., 2018; Chen et al., 2022), physicochemical properties-based split (Kalemati et al., 2024), cold-drug split (Huang et al., 2021), and SIMPD (Landrum et al., 2023), among others.

## 6 CONCLUSION

In this paper, we show the generality of the imbalanced distribution of samples by randomized split and the consequent overshadowing of low-similarity samples. To address the issue, we proposed a novel and flexible similarity aware split strategy for the test set to achieve a desired distribution like uniform discrete distribution, which can deliver a comprehensive evaluation for various drug-target binding affinity prediction algorithms. Furthermore, we utilized the similarity aware split to create a "mimic split", splitting the training set and internal test set by replicating the distribution found in an external test set. Our mimic split consistently aids in selecting the optimal hyper-parameter across various deep-learning methods. In the end, our split strategy enables the generation of distributions with minimum or maximum similarity constraints as required.

## 7 ACKNOWLEDGEMENT

This work is partially supported by the Shanghai Artificial Intelligence Laboratory.

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

# A APPENDIX

## A.1 RELATED WORKS ON DATA SPLITTING

In molecular machine learning, including general QSAR tasks, the challenge of fair predictive evaluation has been a longstanding issue (Cherkasov et al., 2014). While randomized split remains the most commonly used strategy for data splitting, it is not always the optimal choice for evaluating machine learning methods. Consequently, various alternative split strategies have been developed to better evaluate the machine learning methods:

- *Time split* (Sheridan, 2013; Stärk et al., 2022; Guan et al., 2023) is employed for datasets containing temporal information, where the model is trained on historical data and tested on more recent data. It may effectively replicate real-world scenarios, however, a significant number of datasets are devoid of time-specific information. In some situations, when the time span is too large or the data distribution changes significantly over time, the model may struggle to perform well on the test set.

- *Scaffold split* (Bemis & Murcko, 1996) is a technique that splits the dataset based on the structural framework of each sample. It is often leveraged in situations involving out-of-distribution data to provide a measure of generalization capabilities (Stanley et al., 2021; Fang et al., 2022; Zhou et al., 2023; Liu et al., 2024). Because scaffold split does not enforce stratification during the partitioning process, it may result in class imbalance (Yang et al., 2019).

- *Stratified split* - also called stratified random sampling - is a sampling method designed to ensure that each fold of a dataset maintains the same distribution of classes as the entire dataset. It achieves this by first dividing the data into different output strata based on class labels and then executing a random partition with the guaranteeing that the entire label range is encompassed within each set (Krstajic et al., 2014; Wu et al., 2018; Mathai et al., 2020; Chen et al., 2022).

- *Cold-drug split* (Huang et al., 2021) is a method for dividing a dataset into multi-protein prediction tasks, where the dataset is split based on entity types, such as proteins, drugs, or DNAs. The process begins by randomly splitting the dataset into training, validation, and test sets based on one chosen entity type. Subsequently, all data samples associated with a specific entity are assigned to the same set to ensure no overlap across splits, ensuring that there is no overlap of the chosen entity type across the splits.

- *SIMPD split* (Landrum et al., 2023) mimics temporal splits in scenarios where temporal information is not accessible. This approach was developed by observing and analyzing disparities observed between earlier and subsequent samples within the scope of medicinal chemistry projects.

- *Dissimilar split* (Atas Guvenilir & Doğan, 2023) divides a dataset into training and test sets by ensuring that samples in each set are dissimilar. This strategy prevents similar samples from appearing in both sets, thereby increasing the difficulty of predictions.

This challenge is closely intertwined with the broader problem of out-of-distribution (OOD) generalization (Tossou et al., 2024), demonstrating its relevance far beyond the confines of individual tasks such as DTA prediction. In fact, machine learning models tend to perform well when the training set shares a similar distribution with the test set (Leonard & Roy, 2006; Puzyn et al., 2011; Cherkasov et al., 2014). However, previous split strategies often yield test sets with distributions that closely mimic the training set (as shown in Figure A.2). Such alignment between the training and test set distributions can lead to overly optimistic assessments of a model's generalization ability, as it fails to account for scenarios where the model is applied to data with significantly different characteristics. SAE provides an effective solution to this issue by enabling more precise control over data distributions through its ability to capture the similarities between data samples. This approach ensures greater adaptability to a wider range of scenarios.

## A.2 DISCUSSION ABOUT THE APPLICATION OF QSAR SCENARIOS

Quantitative Structure-Activity Relationship (QSAR) modeling is a widely used in silico approach for predicting the biological or chemical properties of molecules (De et al., 2022). Previous studies

on QSAR (Sheridan et al., 2004) have shown that prediction accuracy is highly correlated with the similarity between the molecule being predicted and its closest neighbor in the training set. This observation is similar to patterns found in the DTA prediction task. Therefore, our SAE method can also be extended to QSAR tasks.

For instance, Krishnan et al. (2021) introduced a de novo drug design method that incorporates a pre-trained model alongside transfer learning to generate novel inhibitors targeting the human JAK2 protein. In this approach, transfer learning was utilized to capture the features of the target-related chemical space. If the characteristics of the target-related chemical space—particularly the distribution of the external dataset—are already well understood, our SAE can be applied to replicate this distribution during the splitting of training and test sets, thereby enhancing the overall performance.

Similarly, in the task of protein-protein interaction prediction, improper construction of the data split among training, validation, and test sets can lead to severe data leakage and overly optimistic results (Li et al., 2022). To address this issue, one proposed solution is to divide the test set into three distinct classes (Park & Marcotte, 2012): C1, where test pairs consist of proteins that are both present in the training set; C2, where test pairs involve one protein present in the training set; and C3, where neither protein in the test pair is found in the training set. Notably, the three classes can be viewed as specific cases of our SAE split strategy. Furthermore, the SAE approach can be flexibly applied to constructing test sets with varying levels of difficulty to more effectively evaluate the model's generalization.

### A.3    TIME COMPLEXITY AND SPACE COMPLEXITY ANALYSIS

Given the number of iterations $M$, the number of samples $N$, and the number of bins $K$, we analyze the time complexity of a single iteration in Eq. 13, which involves both forward and backward propagation. During forward propagation, computing $o_k$ involves $O(N \cdot K)$ operations, as it requires iterating over $N$ samples and $K$ bins, with softmax and exponential computations. The computation of $r_i$ is more expensive, requiring $O(N^2)$ operations due to the nested summation over $N$ samples. The regularization term $l_{reg}$ involves a simple summation over $N$, contributing $O(N)$ operations. Combining these, the time complexity of one forward propagation is dominated by the $O(N^2)$ and $O(N \cdot K)$ terms, resulting in $O(N^2 + N \cdot K)$, which simplifies to $O(N^2)$ because $K \ll N$. For backward propagation, the computation of gradients with respect to $w_i$ involves similar operations, which results in the same complexity of $O(N^2)$. Additionally, the process of checking constraints involves $N + 1$ Lagrangian multipliers. The forward and backward propagation for this constraint-checking step each have a complexity of $O(N)$. Combining all of these components, the time complexity of one iteration is $O(N^2)$, and the total time complexity of SAE across all iterations is $O(M \cdot N^2)$. The overall space complexity of SAE is primarily determined by the storage requirements for $s_{ij}$ and the intermediate values needed for computing gradients from $r_i$ to $w_j$. As a result, the space complexity is $O(N^2)$.

Empirically, for the IC50 dataset of target EGFR which contains $N = 4,361$ samples, the desired distribution is a uniform over bins $[1/3, 2/3], (1/3, 2/3], (2/3, 1]$. We set the number of iterations to $M = 20,000$. On a single 3090 GPU, SAE completes the process in approximately 270 seconds, utilizing 2,410 MiB of GPU memory.

We would also like to emphasize that SAE is used in the model development stage and as a result, when the model is developed, it will no longer affect the efficiency for high-throughput inference. Note also that SAE needs only to be performed once for a fixed dataset, meaning that it can be reused by different models as long as they are developed on the same dataset. As a result, the time it takes may be overwhelmed by the time used by the heavy model development. When scaling to large datasets, it should be noted that almost all operations within one iteration are parallelable, and thus it will benefit significantly from more powerful GPU devices and distributed computation.

### A.4    DISCUSSION ON THE APPLICATION OF SAE TO LARGE-SCALE DATASETS

In pharmacompany (private) drug libraries for early drug discovery, there might be 200,000 to $10^6$ samples (Hughes et al., 2011). When scaling to these large datasets, a solution based on the sparse matrix is applicable. Specifically, Figure A.1 shows that the majority of pairwise similarities are low. Suppose the desired distribution is uniform over bins $[1/3, 2/3], (1/3, 2/3], (2/3, 1]$ just as in

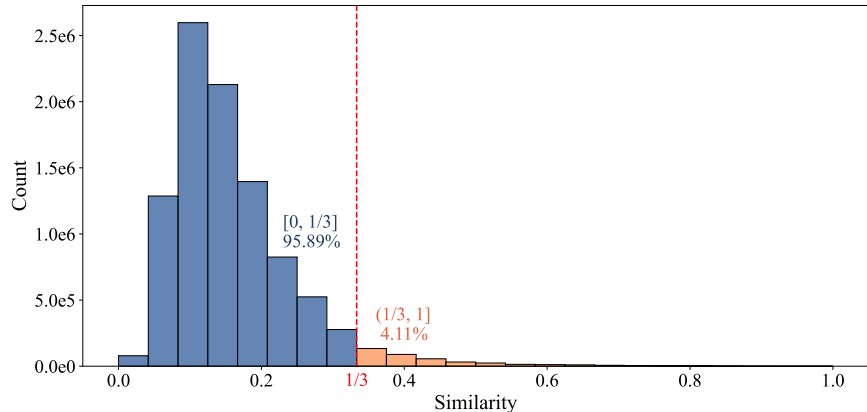

Figure A.1: The similarity distribution of all sample pairs in the IC50 prediction task for the target EGFR. The dataset consists of 4,361 samples, resulting in a total of 9,506,980 pairwise similarity calculations. Among these, 95.89% of the pairs exhibit similarities of no greater than 1/3, while only 4.11% have similarities exceeding 1/3.

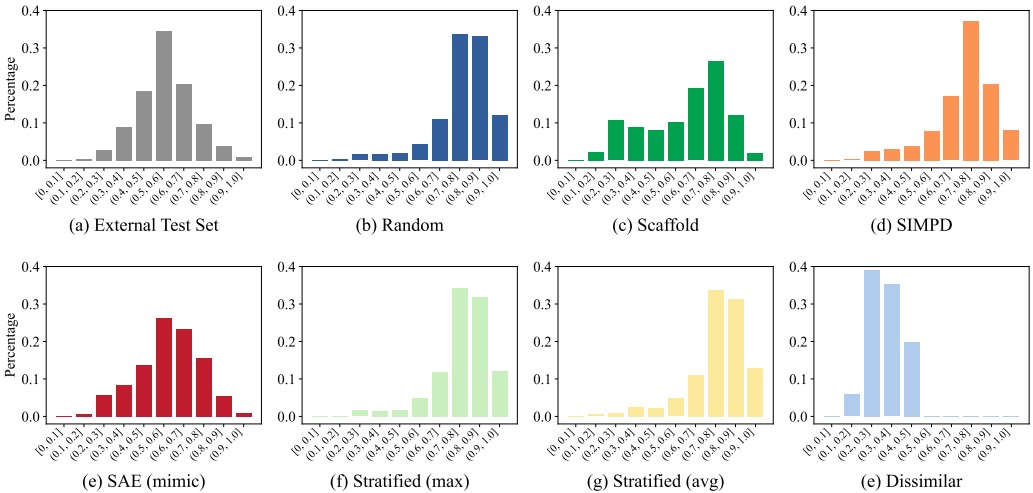

Figure A.2: The similarity distribution of the internal test set across different split strategies. (b) Randomized split leads to a scenario where most internal test samples are highly similar to the training set. (c) Scaffold split produces a more balanced distribution. (d) SIMPD split yields a distribution similar to the random split. (e) Our SAE (mimic) split brings the internal test set's distribution closest to that of the external test set. (f) Stratified split based on the maximum similarity of each ligand to all others in the dataset. (g) Stratified split based on the average similarity of each ligand to all others in the dataset. (h) Dissimilar split guarantees that the similarity will remain below 0.5.

the previous section, over 95% entries in the similarity matrix are less than 1/3 and can be safely set to zero without the interference of the results. The time and space complexity can be significantly reduced in this way. Given $N = 10^6$, the number of pairs with a similarity greater than 1/3 would be approximately $0.05 \cdot N(N-1)/2$, which is about $2.5 \times 10^{10}$. We can store these similarities in a sparse format, represented as tuples (Index of sample A, Index of sample B, Similarity value). Each index can be encoded using 20 bits (sufficient to represent $2^{20} = 1,048,576$ positions), and the similarity value can be quantized into 4 bits (Dettmers et al., 2024). Consequently, the total storage

Table A.1: Comparison of the generalization ability of different split strategies at IC50 for EGFR across five DTA prediction methods.

| Split | Method | Internal Test MAE | Internal Test $R^2$ | External Test MAE | External Test $R^2$ |
|---|---|---|---|---|---|
| Random | FusionDTA | 0.6445 | 0.5399 | 1.1198 | 0.0957 |
| | ChemBERTa | 0.6389 | 0.5574 | 1.0600 | 0.1517 |
| | MolCLR | 0.5830 | 0.6318 | 1.0976 | 0.1546 |
| | PharmHGT | 0.6166 | 0.5739 | 1.1107 | 0.1323 |
| | SAM-DTA | 0.5805 | 0.6301 | 0.9863 | 0.3206 |
| Scaffold | FusionDTA | 0.9626 | 0.1544 | 1.0863 | 0.1243 |
| | ChemBERTa | 0.9314 | 0.1997 | 1.1972 | -0.0491 |
| | MolCLR | 0.8627 | 0.3145 | 1.1585 | 0.0405 |
| | PharmHGT | 0.8537 | 0.3427 | 1.0930 | 0.1594 |
| | SAM-DTA | 0.8725 | 0.3311 | 1.0187 | 0.3034 |
| SIMPD | FusionDTA | 0.7215 | 0.3292 | 1.1417 | 0.0528 |
| | ChemBERTa | 0.6954 | 0.3642 | 1.1010 | 0.0878 |
| | MolCLR | 0.6334 | 0.4775 | 1.2131 | -0.0016 |
| | PharmHGT | 0.6742 | 0.3958 | 1.1588 | 0.0133 |
| | SAM-DTA | 0.6271 | 0.4867 | 1.0058 | 0.3083 |
| Stratified (max) | FusionDTA | 0.6753 | 0.5346 | 1.0886 | 0.1517 |
| | ChemBERTa | 0.6752 | 0.5384 | 1.1504 | 0.0223 |
| | MolCLR | 0.5968 | 0.6504 | 1.0917 | 0.1207 |
| | PharmHGT | 0.6092 | 0.6302 | 1.0694 | 0.1811 |
| | SAM-DTA | 0.6019 | 0.6404 | 1.0345 | 0.2722 |
| Stratified (avg) | FusionDTA | 0.6490 | 0.5713 | 1.0957 | 0.1206 |
| | ChemBERTa | 0.6724 | 0.5191 | 1.1258 | 0.0896 |
| | MolCLR | 0.5939 | 0.6368 | 1.1556 | 0.1019 |
| | PharmHGT | 0.6103 | 0.5895 | 1.0938 | 0.1667 |
| | SAM-DTA | 0.6099 | 0.6159 | 0.9946 | 0.3345 |
| Dissimilar | FusionDTA | 0.9425 | -0.1256 | 1.2788 | -0.1063 |
| | ChemBERTa | 0.8927 | -0.0139 | 1.6402 | -0.5971 |
| | MolCLR | 0.8462 | 0.0592 | 1.3355 | -0.1366 |
| | PharmHGT | 0.9011 | -0.0029 | 1.6006 | -0.5237 |
| | SAM-DTA | 0.9239 | -0.0845 | 1.2140 | -0.0039 |
| SAE (mimic) | FusionDTA | 0.9130 | 0.2919 | 1.0605 | 0.2122 |
| | ChemBERTa | 0.8976 | 0.2736 | 1.0452 | 0.2477 |
| | MolCLR | 0.8536 | 0.3653 | 1.0002 | 0.2981 |
| | PharmHGT | 0.8826 | 0.3200 | 1.0609 | 0.1861 |
| | SAM-DTA | 0.8545 | 0.3770 | 0.9773 | 0.3367 |

requirement can be calculated as:

$$(20bits + 20bits + 4bits) \times 2.5 \times 10^{10} = 5.5 Bytes \times 2.5 \times 10^{10} \approx 128GB$$

This size is manageable and can even be stored in memory.

## A.5 SUPPLEMENTARY EXPERIMENTAL RESULTS OF MIMIC SPLIT

For thorough comparison with other split strategies, we implemented stratified split (Wu et al., 2018; Chen et al., 2022) and dissimilar split (Atas Guvenilir & Doğan, 2023) at IC50 for EGFR. For the stratified split, we first compute the pairwise similarities for the full dataset, resulting in a similarity matrix of size $N \times N$ (where $N$ is the number of samples in the dataset, with the diagonal values set to zero). Next, we calculate the maximum/average similarity for each row, yielding a similarity vector of size $N$, which represents the maximum/average similarity of each ligand to all others in

Table A.2: Detailed comparison of the generalization ability of different split strategies at IC50 for EGFR across five DTA prediction methods.

| Bin | Count | Split | FusionDTA | ChemBERTa | MolCLR | PharmHGT | SAM-DTA |
|---|---|---|---|---|---|---|---|
| | | | Extrenal Test MAE | | | | |
| [0, 1/3] | 120 | Random | 1.2442 | 1.1167 | 1.2385 | 1.3261 | 1.1343 |
| | | Scaffold | 1.2293 | 1.2242 | 1.1630 | 1.1840 | 1.1363 |
| | | SIMPD | 1.4297 | 1.3223 | 1.7376 | 1.3382 | 1.1179 |
| | | Stratified (max) | 1.2311 | 1.1663 | 1.2414 | 1.2179 | 1.1747 |
| | | Stratified (avg) | 1.5371 | 1.4478 | 1.6005 | 1.3962 | 1.3666 |
| | | Dissimilar | 1.4134 | 1.2363 | 1.1912 | 1.2424 | 1.4161 |
| | | SAE (mimic) | 1.0626 | 1.0315 | 1.1848 | 1.1082 | 1.0435 |
| (1/3, 2/3] | 1026 | Random | 1.1416 | 1.0874 | 1.0983 | 1.0989 | 0.9879 |
| | | Scaffold | 1.0729 | 1.2273 | 1.2021 | 1.0891 | 1.0209 |
| | | SIMPD | 1.1545 | 1.1272 | 1.2134 | 1.1614 | 1.0158 |
| | | Stratified (max) | 1.0756 | 1.1611 | 1.0927 | 1.0677 | 1.0317 |
| | | Stratified (avg) | 1.0815 | 1.1450 | 1.1384 | 1.0835 | 0.9808 |
| | | Dissimilar | 1.3106 | 1.6954 | 1.3676 | 1.6504 | 1.2513 |
| | | SAE (mimic) | 1.0531 | 1.0594 | 0.9979 | 1.0606 | 0.9743 |
| (2/3, 1] | 186 | Random | 0.9559 | 0.9060 | 1.0208 | 1.0516 | 0.9015 |
| | | Scaffold | 1.0604 | 1.0346 | 0.9449 | 1.0543 | 0.9335 |
| | | SIMPD | 0.9741 | 0.9022 | 1.0066 | 1.0775 | 0.9191 |
| | | Stratified (max) | 1.0718 | 1.0883 | 1.0017 | 0.9933 | 0.9683 |
| | | Stratified (avg) | 0.9694 | 0.9047 | 1.0415 | 1.0102 | 0.8967 |
| | | Dissimilar | 1.0201 | 1.6720 | 1.2858 | 1.6248 | 0.8773 |
| | | SAE (mimic) | 1.1003 | 0.9762 | 0.8938 | 1.0322 | 0.9511 |
| | | | External Test $R^2$ | | | | |
| [0, 1/3] | 120 | Random | -0.5461 | -0.5827 | -0.5009 | -0.6605 | -0.2855 |
| | | Scaffold | -0.1335 | -0.3903 | -0.1887 | -0.1096 | -0.0121 |
| | | SIMPD | -0.8717 | -0.7917 | -1.3400 | -0.6245 | -0.1181 |
| | | Stratified (max) | -0.4278 | -0.4195 | -0.7345 | -0.3873 | -0.3106 |
| | | Stratified (avg) | -0.8887 | -1.0024 | -1.1185 | -0.5790 | -0.3962 |
| | | Dissimilar | -0.4209 | -0.0067 | 0.0122 | -0.0173 | -0.4122 |
| | | SAE (mimic) | -0.0736 | -0.1140 | -0.3105 | -0.2592 | -0.0783 |
| (1/3, 2/3] | 1026 | Random | 0.0351 | 0.1066 | 0.1433 | 0.1005 | 0.2950 |
| | | Scaffold | 0.0987 | -0.1211 | -0.0502 | 0.1355 | 0.2824 |
| | | SIMPD | 0.0078 | 0.0332 | -0.0169 | -0.0337 | 0.2780 |
| | | Stratified (max) | 0.1305 | -0.0321 | 0.1016 | 0.1530 | 0.2633 |
| | | Stratified (avg) | 0.1017 | 0.0476 | 0.1156 | 0.1430 | 0.3188 |
| | | Dissimilar | -0.2307 | -0.8279 | -0.2834 | -0.7374 | -0.1244 |
| | | SAE (mimic) | 0.1791 | 0.1848 | 0.2632 | 0.1237 | 0.3020 |
| (2/3, 1] | 186 | Random | 0.0869 | 0.1390 | -0.0844 | 0.0577 | 0.2594 |
| | | Scaffold | -0.0967 | -0.0351 | 0.1475 | -0.0412 | 0.2110 |
| | | SIMPD | 0.1642 | 0.2476 | 0.0842 | 0.0139 | 0.2880 |
| | | Stratified (max) | 0.0272 | -0.0595 | 0.1063 | 0.1111 | 0.1550 |
| | | Stratified (avg) | 0.1114 | 0.2234 | 0.0081 | 0.0860 | 0.3187 |
| | | Dissimilar | 0.0288 | -1.0679 | -0.3223 | -0.9576 | 0.2743 |
| | | SAE (mimic) | -0.2135 | 0.1123 | 0.1807 | 0.0350 | 0.1359 |

the dataset. Finally, we divide the dataset into K bins based on the maximal/average similarity and perform random sampling within each bin to create the test set. We refer to the two variations of this stratified split strategy as "Stratified (max)," which uses the maximum similarity for binning,

and "Stratified (avg)", which uses the average similarity. The similarity distributions of the stratified split are shown in Figure A.2 (f) and Figure A.2 (g). The distribution result of dissimilar split is shown in Figure A.2 (h). A comparison of the generalization ability of different split strategies is shown in Table A.1. SAE performs better than stratified sampling and dissimilar split with a clear margin.

We also analyze the performance of the different brackets in the external dataset. As is shown in Table A.2, SAE improves performance in the low- and mid-similarity brackets, but not in the high-similarity one. We believe this is because the internal test set by SAE has more samples in the low- and mid-similarity brackets and thus performance in these brackets receives more attention compared with other split strategies.

### A.6    COMPARISON ACROSS DIFFERENT SIMILARITY MEASURES AND FINGERPRINTS

As for comparison across different similarity measures and fingerprints, we conducted experiments on similarity measure choices including Tanimoto, Cosine, Sokal, and Dice, and fingerprint choices including Morgan (ECFP), RDKFP (RDKit), and Avalon. As shown by Table A.3, split results are less affected by similarity measure choices but more influenced by fingerprint choices. In all settings of similarity measures and fingerprints, SAE outperforms other approaches by achieving a split that is closer to the desired distribution (uniform distribution in this case).

Table A.3: Comparison across different similarity measures and fingerprints, the desired distribution is a uniform distribution across the bins [0, 1/3], (1/3, 2/3], and (2/3, 1]. The numbers in this table are presented in the form [Sample counts in the first bin, Sample counts in the second bin, Sample counts in the third bin].

| Similarity Measure | Fingerprint | SAE (balanced) | Random | Scaffold | SIMPD | Stratified (max) | Stratified (avg) |
|---|---|---|---|---|---|---|---|
| Cosine | Morgan | 145, 436, 292 | 0, 33, 840 | 6, 159, 708 | 16, 657, 200 | 0, 32, 841 | 0, 27, 846 |
| | RDKFP | 18, 426, 429 | 0, 17, 856 | 1, 78, 794 | 1, 124, 748 | 0, 13, 860 | 1, 14, 858 |
| | Avalon | 9, 429, 435 | 0, 7, 866 | 0, 21, 852 | 0, 16, 857 | 0, 6, 867 | 0, 6, 867 |
| Sokal | Morgan | 292, 289, 292 | 33, 398, 442 | 172, 510, 191 | 689, 126, 58 | 34, 423, 416 | 29, 416, 428 |
| | RDKFP | 291, 291, 291 | 19, 80, 774 | 85, 236, 552 | 135, 624, 114 | 14, 82, 777 | 15, 74, 784 |
| | Avalon | 291, 291, 291 | 7, 63, 803 | 26, 275, 572 | 23, 629, 221 | 8, 76, 789 | 6, 73, 794 |
| Dice | Morgan | 182, 378, 313 | 0, 33, 840 | 9, 163, 701 | 17, 672, 184 | 0, 34, 839 | 2, 27, 844 |
| | RDKFP | 60, 463, 350 | 0, 19, 854 | 2, 83, 788 | 1, 134, 738 | 0, 14, 859 | 1, 14, 858 |
| | Avalon | 32, 416, 425 | 0, 7, 866 | 0, 26, 847 | 0, 23, 850 | 0, 8, 865 | 0, 6, 867 |
| Tanimoto | Morgan | 290, 299, 284 | 16, 98, 759 | 80, 273, 520 | 228, 547, 98 | 12, 99, 762 | 13, 99, 761 |
| | RDKFP | 289, 292, 292 | 8, 34, 831 | 15, 184, 674 | 2, 635, 236 | 1, 39, 833 | 4, 33, 836 |
| | Avalon | 220, 325, 328 | 0, 28, 845 | 2, 154, 717 | 1, 324, 548 | 0, 30, 843 | 1, 28, 844 |

Table A.4: Hyper-parameters used in the mimic split experiment for each method, the following search options are derived from the default parameter settings of each method.

| Method | Hyper-parameter | Options |
|---|---|---|
| SAM-DTA | Optimizer | [Adam, SGD] |
| | Learning rate | [1e-3, 1e-4, 1e-5] |
| | Batch size | [10, 32, 64] |
| MolCLR | Optimizer | [Adam, SGD] |
| | Learning rate (prediction head, GNN encoder) | [(1e-3, 5e-3), (1e-4, 5e-4), (1e-5, 5e-5)] |
| | Dropout ratio | [0.3, 0.5] |
| | Readout pooling | [Mean, Max, Add] |
| FusionDTA | Optimizer | [Adam, SGD] |
| | Learning rate | [1e-2, 1e-3, 1e-4] |
| | Batch size | [128, 256] |
| | Loss function | [L1, MSE] |
| PharmHGT | Optimizer | [Adam, SGD] |
| | Learning rate | [1e-2, 1e-3, 1e-4] |
| | Activation function | [Sigmoid, ReLU] |
| | Loss function | [RMSE, MAE] |
| ChemBERTa | Optimizer | [AdamW, Adafactor] |
| | Learning rate | [4e-3, 4e-4, 4e-5, 4e-6] |
| | Batch size | [4, 8, 16] |

Table A.5: Variations of $SimilarityToTrainingSet$ related to feature extraction, similarity measure, aggregation functions, and performance metrics. The methods for the detailed showcase are FusionDTA and ChemBERTa.

| | Randomized Split (MAE) | | | | | |
|---|---|---|---|---|---|---|
| Bin | Feature: RDKit fingerprint | | | Feature: Avalon fingerprint | | |
| | Count (Ratio) | FusionDTA | ChemBERTa | Count (Ratio) | FusionDTA | ChemBERTa |
| [0 , 1/3] | 8 (0.0092) | 1.4442 | 1.4679 | 0 (0.0000) | - | - |
| (1/3, 2/3] | 34 (0.0389) | 1.1294 | 1.1306 | 28 (0.0321) | 1.3299 | 1.2340 |
| (2/3, 1] | 831 (0.9519) | 0.6407 | 0.6546 | 845 (0.9679) | 0.6451 | 0.6623 |
| overall | 873 (1.0000) | 0.6671 | 0.6806 | 873 (1.0000) | 0.6671 | 0.6806 |
| | SimilarityMeasure: Sokal similarity | | | SimilarityMeasure: Dice coefficient | | |
| [0 , 1/3] | 33 (0.0378) | 1.2751 | 1.2711 | 0 (0.0000) | - | - |
| (1/3, 2/3] | 398 (0.4559) | 0.7366 | 0.7234 | 33 (0.0378) | 1.2751 | 1.2711 |
| (2/3, 1] | 442 (0.5063) | 0.5591 | 0.5980 | 840 (0.9622) | 0.6432 | 0.6574 |
| overall | 873 (1.0000) | 0.6671 | 0.6806 | 873 (1.0000) | 0.6671 | 0.6806 |
| | Aggregation: Top-3 | | | Aggregation: Top-5 | | |
| [0 , 1/3] | 17 (0.0195) | 1.1567 | 1.3484 | 24 (0.0275) | 1.3682 | 1.3330 |
| (1/3, 2/3] | 171 (0.1959) | 0.8839 | 0.8722 | 240 (0.2749) | 0.7861 | 0.8094 |
| (2/3, 1] | 685 (0.7847) | 0.6008 | 0.6162 | 609 (0.6976) | 0.5926 | 0.6041 |
| overall | 873 (1.0000) | 0.6671 | 0.6806 | 873 (1.0000) | 0.6671 | 0.6806 |
| | Randomized Split ($R^2$) | | | | | |
| Bin | Feature: RDKit fingerprint | | | Feature: Avalon fingerprint | | |
| | Count (Ratio) | FusionDTA | ChemBERTa | Count (Ratio) | FusionDTA | ChemBERTa |
| [0 , 1/3] | 8 (0.0092) | 0.2319 | 0.2536 | 0 (0.0000) | - | - |
| (1/3, 2/3] | 34 (0.0389) | 0.2253 | 0.1829 | 28 (0.0321) | 0.2166 | 0.2573 |
| (2/3, 1] | 831 (0.9519) | 0.5899 | 0.5796 | 845 (0.9679) | 0.5845 | 0.5698 |
| overall | 873 (1.0000) | 0.5697 | 0.5585 | 873 (1.0000) | 0.5697 | 0.5585 |
| | SimilarityMeasure: Sokal similarity | | | SimilarityMeasure: Dice coefficient | | |
| [0 , 1/3] | 33 (0.0378) | 0.0871 | 0.0615 | 0 (0.0000) | - | - |
| (1/3, 2/3] | 398 (0.4559) | 0.5068 | 0.5186 | 33 (0.0378) | 0.0871 | 0.0615 |
| (2/3, 1] | 442 (0.5063) | 0.6469 | 0.6117 | 840 (0.9622) | 0.5898 | 0.5793 |
| overall | 873 (1.0000) | 0.5697 | 0.5585 | 873 (1.0000) | 0.5697 | 0.5585 |
| | Aggregation: Top-3 | | | Aggregation: Top-5 | | |
| [0 , 1/3] | 17 (0.0195) | -0.0685 | -0.3530 | 24 (0.0275) | -0.0237 | -0.0345 |
| (1/3, 2/3] | 171 (0.1959) | 0.4390 | 0.4408 | 240 (0.2749) | 0.4881 | 0.4647 |
| (2/3, 1] | 685 (0.7847) | 0.5930 | 0.5837 | 609 (0.6976) | 0.5965 | 0.5903 |
| overall | 873 (1.0000) | 0.5697 | 0.5585 | 873 (1.0000) | 0.5697 | 0.5585 |

Table A.6: Variations of $SimilarityToTrainingSet$ related to feature extraction, similarity measure, aggregation functions, and performance metrics. The method for the detailed showcase is MolCLR.

| Randomized Split (MAE) | | | | |
|---|---|---|---|---|
| Bin | Feature: RDKit fingerprint | | Feature: Avalon fingerprint | |
| | Count (Ratio) | MolCLR | Count (Ratio) | MolCLR |
| [0 , 1/3] | 8 (0.0092) | 1.4442 | 0 (0.0000) | - |
| (1/3, 2/3] | 34 (0.0389) | 1.1294 | 28 (0.0321) | 1.3299 |
| (2/3, 1] | 831 (0.9519) | 0.6407 | 845 (0.9679) | 0.6451 |
| overall | 873 (1.0000) | 0.6671 | 873 (1.0000) | 0.6671 |
| | SimilarityMeasure: Sokal similarity | | SimilarityMeasure: Dice coefficient | |
| [0 , 1/3] | 33 (0.0378) | 1.2751 | 0 (0.0000) | - |
| (1/3, 2/3] | 398 (0.4559) | 0.7366 | 33 (0.0378) | 1.2751 |
| (2/3, 1] | 442 (0.5063) | 0.5591 | 840 (0.9622) | 0.6432 |
| overall | 873 (1.0000) | 0.6671 | 873 (1.0000) | 0.6671 |
| | Aggregation: Top-3 | | Aggregation: Top-5 | |
| [0 , 1/3] | 17 (0.0195) | 1.1567 | 24 (0.0275) | 1.3682 |
| (1/3, 2/3] | 171 (0.1959) | 0.8839 | 240 (0.2749) | 0.7861 |
| (2/3, 1] | 685 (0.7847) | 0.6008 | 609 (0.6976) | 0.5926 |
| overall | 873 (1.0000) | 0.6671 | 873 (1.0000) | 0.6671 |
| Randomized Split ($R^2$) | | | | |
| Bin | Feature: RDKit fingerprint | | Feature: Avalon fingerprint | |
| | Count (Ratio) | MolCLR | Count (Ratio) | MolCLR |
| [0 , 1/3] | 8 (0.0092) | 0.2319 | 0 (0.0000) | - |
| (1/3, 2/3] | 34 (0.0389) | 0.2253 | 28 (0.0321) | 0.2166 |
| (2/3, 1] | 831 (0.9519) | 0.5899 | 845 (0.9679) | 0.5845 |
| overall | 873 (1.0000) | 0.5697 | 873 (1.0000) | 0.5697 |
| | SimilarityMeasure: Sokal similarity | | SimilarityMeasure: Dice coefficient | |
| [0 , 1/3] | 33 (0.0378) | 0.0871 | 0 (0.0000) | - |
| (1/3, 2/3] | 398 (0.4559) | 0.5068 | 33 (0.0378) | 0.0871 |
| (2/3, 1] | 442 (0.5063) | 0.6469 | 840 (0.9622) | 0.5898 |
| overall | 873 (1.0000) | 0.5697 | 873 (1.0000) | 0.5697 |
| | Aggregation: Top-3 | | Aggregation: Top-5 | |
| [0 , 1/3] | 17 (0.0195) | -0.0685 | 24 (0.0275) | -0.0237 |
| (1/3, 2/3] | 171 (0.1959) | 0.4390 | 240 (0.2749) | 0.4881 |
| (2/3, 1] | 685 (0.7847) | 0.5930 | 609 (0.6976) | 0.5965 |
| overall | 873 (1.0000) | 0.5697 | 873 (1.0000) | 0.5697 |

Table A.7: Comparison of Randomized Split and SAE (balanced) Split at IC50 for BACE1, Ki for Carbonic anhydrase I and Carbonic anhydrase II. The methods for the detailed showcase are FusionDTA and ChemBERTa.

| | IC50 for Target BACE1 (MAE) | | | | | |
|---|---|---|---|---|---|---|
| Bin | Randomized Split | | | SAE (balanced) Split | | |
| | Count (Ratio) | FusionDTA | ChemBERTa | Count (Ratio) | FusionDTA | ChemBERTa |
| [0 , 1/3] | 10 (0.0108) | 1.3020 | 1.1440 | 309 (0.3330) | 1.2117 | 1.2503 |
| (1/3, 2/3] | 67 (0.0722) | 0.7270 | 0.6719 | 311 (0.3351) | 0.7444 | 0.6599 |
| (2/3, 1] | 851 (0.9170) | 0.5105 | 0.5267 | 308 (0.3319) | 0.5310 | 0.5352 |
| overall | 928 (1.0000) | 0.5347 | 0.5439 | 928 (1.0000) | 0.8292 | 0.8151 |
| | IC50 for Target BACE1 ($R^2$) | | | | | |
| [0 , 1/3] | 10 (0.0108) | 0.0422 | 0.3204 | 309 (0.3330) | -0.5113 | -0.5641 |
| (1/3, 2/3] | 67 (0.0722) | 0.5980 | 0.6325 | 311 (0.3351) | 0.4238 | 0.5787 |
| (2/3, 1] | 851 (0.9170) | 0.6651 | 0.6446 | 308 (0.3319) | 0.7076 | 0.7235 |
| overall | 928 (1.0000) | 0.6755 | 0.6673 | 928 (1.0000) | 0.4213 | 0.4548 |
| | Ki for Target Carbonic anhydrase I (MAE) | | | | | |
| Bin | Randomized Split | | | SAE (balanced) Split | | |
| | Count (Ratio) | FusionDTA | ChemBERTa | Count (Ratio) | FusionDTA | ChemBERTa |
| [0 , 1/3] | 7 (0.0079) | 0.9363 | 0.9564 | 264 (0.2983) | 1.0252 | 0.9245 |
| (1/3, 2/3] | 205 (0.2316) | 0.7086 | 0.7085 | 311 (0.3514) | 0.7362 | 0.7228 |
| (2/3, 1] | 673 (0.7605) | 0.5203 | 0.5440 | 310 (0.3503) | 0.6181 | 0.6060 |
| overall | 885 (1.0000) | 0.5673 | 0.5854 | 885 (1.0000) | 0.7810 | 0.7421 |
| | Ki for Target Carbonic anhydrase I ($R^2$) | | | | | |
| [0 , 1/3] | 7 (0.0079) | 0.0634 | 0.1421 | 264 (0.2983) | -0.4131 | -0.1076 |
| (1/3, 2/3] | 205 (0.2316) | 0.3536 | 0.3761 | 311 (0.3514) | 0.2106 | 0.2829 |
| (2/3, 1] | 673 (0.7605) | 0.4500 | 0.4231 | 310 (0.3503) | 0.3532 | 0.3299 |
| overall | 885 (1.0000) | 0.4259 | 0.4161 | 885 (1.0000) | 0.1253 | 0.2334 |
| | Ki for Target Carbonic anhydrase II (MAE) | | | | | |
| Bin | Randomized Split | | | SAE (balanced) Split | | |
| | Count (Ratio) | FusionDTA | ChemBERTa | Count (Ratio) | FusionDTA | ChemBERTa |
| [0 , 1/3] | 8 (0.0087) | 0.8465 | 0.5778 | 244 (0.2667) | 0.9849 | 0.9314 |
| (1/3, 2/3] | 201 (0.2197) | 0.6817 | 0.7419 | 342 (0.3738) | 0.8265 | 0.7390 |
| (2/3, 1] | 706 (0.7716) | 0.5605 | 0.5997 | 329 (0.3596) | 0.6040 | 0.6072 |
| overall | 915 (1.0000) | 0.5896 | 0.6307 | 915 (1.0000) | 0.7888 | 0.7429 |
| | Ki for Target Carbonic anhydrase II ($R^2$) | | | | | |
| [0 , 1/3] | 8 (0.0087) | 0.0349 | 0.3581 | 244 (0.2667) | 0.0488 | 0.2523 |
| (1/3, 2/3] | 201 (0.2197) | 0.5603 | 0.4667 | 342 (0.3738) | 0.3416 | 0.4686 |
| (2/3, 1] | 706 (0.7716) | 0.5513 | 0.5146 | 329 (0.3596) | 0.4499 | 0.4744 |
| overall | 915 (1.0000) | 0.5570 | 0.5087 | 915 (1.0000) | 0.3583 | 0.4659 |

Table A.8: Comparison of Randomized Split and SAE (balanced) Split at IC50 for BACE1, Ki for Carbonic anhydrase I and Carbonic anhydrase II. The method for the detailed showcase is MolCLR.

| | IC50 for Target BACE1 (MAE) | | | |
|---|---|---|---|---|
| Bin | Randomized Split | | SAE (balanced) Split | |
| | Count (Ratio) | MolCLR | Count (Ratio) | MolCLR |
| [0  , 1/3] | 10 (0.0108) | 1.2452 | 309 (0.3330) | 1.4141 |
| (1/3, 2/3] | 67 (0.0722) | 0.6952 | 311 (0.3351) | 0.6940 |
| (2/3,  1] | 851 (0.9170) | 0.4878 | 308 (0.3319) | 0.4784 |
| overall | 928 (1.0000) | 0.5109 | 928 (1.0000) | 0.8622 |
| | IC50 for Target BACE1 ($R^2$) | | | |
| [0  , 1/3] | 10 (0.0108) | -0.0492 | 309 (0.3330) | -0.9211 |
| (1/3, 2/3] | 67 (0.0722) | 0.6184 | 311 (0.3351) | 0.5107 |
| (2/3,  1] | 851 (0.9170) | 0.6919 | 308 (0.3319) | 0.7746 |
| overall | 928 (1.0000) | 0.6974 | 928 (1.0000) | 0.3713 |
| | Ki for Target Carbonic anhydrase I (MAE) | | | |
| Bin | Randomized Split | | SAE (balanced) Split | |
| | Count (Ratio) | MolCLR | Count (Ratio) | MolCLR |
| [0  , 1/3] | 7 (0.0079) | 0.8510 | 264 (0.2983) | 1.0059 |
| (1/3, 2/3] | 205 (0.2316) | 0.6141 | 311 (0.3514) | 0.7549 |
| (2/3,  1] | 673 (0.7605) | 0.4762 | 310 (0.3503) | 0.5755 |
| overall | 885 (1.0000) | 0.5111 | 885 (1.0000) | 0.7669 |
| | Ki for Target Carbonic anhydrase I ($R^2$) | | | |
| [0  , 1/3] | 7 (0.0079) | 0.3127 | 264 (0.2983) | -0.3331 |
| (1/3, 2/3] | 205 (0.2316) | 0.5338 | 311 (0.3514) | 0.1585 |
| (2/3,  1] | 673 (0.7605) | 0.5598 | 310 (0.3503) | 0.4342 |
| overall | 885 (1.0000) | 0.5572 | 885 (1.0000) | 0.1552 |
| | Ki for Target Carbonic anhydrase II (MAE) | | | |
| Bin | Randomized Split | | SAE (balanced) Split | |
| | Count (Ratio) | MolCLR | Count (Ratio) | MolCLR |
| [0  , 1/3] | 8 (0.0087) | 1.0278 | 244 (0.2667) | 0.8873 |
| (1/3, 2/3] | 201 (0.2197) | 0.6882 | 342 (0.3738) | 0.6907 |
| (2/3,  1] | 706 (0.7716) | 0.5497 | 329 (0.3596) | 0.6232 |
| overall | 915 (1.0000) | 0.5843 | 915 (1.0000) | 0.7189 |
| | Ki for Target Carbonic anhydrase II ($R^2$) | | | |
| [0  , 1/3] | 8 (0.0087) | -0.3461 | 244 (0.2667) | 0.2192 |
| (1/3, 2/3] | 201 (0.2197) | 0.5635 | 342 (0.3738) | 0.5347 |
| (2/3,  1] | 706 (0.7716) | 0.5964 | 329 (0.3596) | 0.4409 |
| overall | 915 (1.0000) | 0.5856 | 915 (1.0000) | 0.4740 |

