# OpenReview forum: "Rethinking the generalization of drug target affinity prediction algorithms via similarity aware evaluation"
_ICLR.cc/2025/Conference — ICLR 2025 Oral_

### Official Review · Reviewer_MUd6 · 2024-10-23

**Soundness:** 4
**Presentation:** 3
**Contribution:** 3
**Rating:** 8
**Confidence:** 4

**Summary:**

This paper studies the problem of generalization in the problem of predicting of drug target binding for drug repurposing, where the affinity score to a specific target is computed on a library of drugs. This paper first draws attention through empirical results to the fact that good performance on a testing set which low-similarity points are underrepresented might be misleading, as the trained model might fail in a real-life setting. Second, the paper introduces an approach to splitting the training and testing subsets for the training of models, that aims at achieving a target distribution on points in the testing set, that might be less biased for the evaluation of a prediction model.

**Strengths:**

- Clarity: The submission is experimentally rigorous, and well-written.
- Quality: Several interesting claims are made and illustrated by relevant experiments (for instance, that performance on the testing set is correlated with the proportion of low-similarity samples and that those are often a small part of the dataset). The tested baselines are relevant, and the impact of all relevant hyperparameters is appropriately assessed.
- Significance: The question is important, in particular for healthcare applications, and the algorithmic contribution is valuable, as it is a tractable approach (optimization-wise).
- Originality: The algorithm proposed to splitting data can handle any target distribution.

**Weaknesses:**

- Clarity: The section on related works should be present earlier in the paper.
- Quality: The results shown are not easily reproducible, as the code is not shared. Some aspects to make the method truly appealing to the end users (training for prediction models) are missing: the computational cost of the approach (the computational resources needed to run the experiments), the empirical runtime on the data set of approximately 4,000 samples for instance, and how it compares for these aspects to the selected baselines (random, scaffold, SIMPD). The impact of the number of bins K on the closeness between solution distribution and target distribution is not discussed.
- Significance: There seems to be a significant gap between the target distribution and the one returned at the end (Figure 6 : 20% of points are present in bins that should remain empty), which might be due to the several approximations made to obtain a tractable optimization problem. I assume that the approach to obtain this figure is to set to zero the number of elements expected in bins associated with similarity ranges higher than the prescribed threshold, but this is not specified in the paper. Finally, the approach relies on the precomputation of a similarity matrix that might be too large to be stored in-place. All the data sets in the paper are of size less than 5,500, but in pharmacompany (private) drug libraries for early drug discovery, there might be 200,000 to $10^6$ compounds [1].

[1] Hughes, James P., et al. "Principles of early drug discovery." British journal of pharmacology 162.6 (2011): 1239-1249.

Even though I rated the paper 8, I think the reproducibility is the largest issue and the absence of code is not justified in the paper. This is a major concern to me.

**Questions:**

- Could you provide the code for your method or explain why it is not included? Having access to the implementation would greatly aid reproducibility.
- People often split the data into three subsets: training and validation sets (to apply the method on the training set and use the metrics computed on the validation set to update parameters) and testing set (to evaluate the performance of the method). In that case, should the SAE method be applied iteratively? As it is, it can only split data into two groups.
- For Figure 6, could you specify the exact target distribution used to generate these plots, including any constraints or parameters?

Summary of suggested modifications:
- Please provide experimental code or justify why it is not included.
- Mention how the approach could be adapted for three-way splits, or what modifications would be needed to support this common scenario.

---

> ### Author Response · Authors · 2024-11-23
>
> Thank you for your comments. The code will surely be released upon publication to facilitate the reproducibility and future research. With respect to the scenario of three-way splits, since SAE can be used to split testset from the union of trainset and valset (denoted as trainset+valset), the problem of three-way splits is essentially how to split valset from trainset. Note that valset is used for model selection and it should be close in distribution to the testset. Applying SAE iteratively, as you have mentioned, is a simple and effective way. There is only a subtle issue that similarity measured is slightly different in the two rounds of the iteration: similarity is measured on testset from trainset+valset in the first round but on valset from trainset in the second round. To address it, one can locate samples in trainset+valset which achieve the maximum in Eq.(5) of the first round and put them directly in the trainset in the second round. In this way, similarity on testset from trainset+valset will be the same as from trainset only.
>
> For Figure 6, the target distribution is uniform over the specified similarity range. Specifically, for (a) the target distribution is uniform over similarity bins [0, 0.1], (0.1, 0.2], (0.2, 0.3] and (0.3, 0.4]; for (b) it is uniform over similarity bins [0, 0.1], (0.1, 0.2], (0.2, 0.3], (0.3, 0.4], (0.4, 0.5] and (0.5, 0.6]; for (c)
> it is uniform over similarity bins [0.4, 0.5] and (0.5, 0.6].
>
> As for the time and space complexity, for the EGFR dataset in the paper which contains 4,361 ligands, with the number of iterations set to 20,000, SAE takes approximately 270 seconds on a single NVIDIA GTX 3090 GPU, with a memory usage of 2,410 MiB. When scaling to large datasets, for example 10^6 compounds as you have mentioned, a solution based on sparse matrix is applicable. Specifically, in Figure A.1 of the appendix we analyze the distribution of pairwise similarity and find that the majority is low. Suppose the desired distribution is uniform over bins [0, 1/3], (1/3, 2/3] and (2/3, 1] just as in the paper, over 95% entries in the similarity matrix are less than 1/3 and can be safely set to zero without interference of the results. The time and space complexity can be significantly reduced in this way. We appreciate that you point out the missing analysis of the time and space complexity. It is important to practitioners and of high relevance to the scalability. These discussions have been added in the Section A.3 and Section A.4 of the revised version of the paper.

---

> > ### Comment · Reviewer_MUd6 · 2024-11-25
> >
> > Thanks for your response. The answers are convincing and the added sections on complexity are welcome. I keep my score as it is, as it is a paper worthy of acceptance to me.

---

### Official Review · Reviewer_PwZH · 2024-10-31

**Soundness:** 3
**Presentation:** 4
**Contribution:** 4
**Rating:** 8
**Confidence:** 5

**Summary:**

This paper targets the performance evaluation of drug-target binding affinity prediction, focusing on analyzing effective data partitioning into training and validation sets. The paper proposes a similarity-aware evaluation (SAE) as an alternative to the conventional random split approach. Recognizing that the similarity of test set molecules to those in the training set significantly impacts machine learning prediction performance, the paper first categorizes validation set molecules into three groups based on their similarity scores with training set molecules. Through extensive experimentation, the paper demonstrates (1) that with random splitting, a large portion of the validation set is occupied by molecules similar to those in the training set, regardless of the feature representation used, and (2) that the overall prediction accuracy aligns closely with the prediction accuracy for training-set-similar molecules dominating the validation set.

The paper argues that, in real-world external test scenarios, there is no inherent focus on molecules similar to those in the training set, and hence, an evaluation approach that is independent of similarity distribution is preferable. However, this cannot be achieved through traditional random splitting. The paper then introduces SAE as a sampling strategy that considers similarity bins and optimizes the scaled distance between the similarity distribution within each bin and a uniform distribution, achieving combinatorial optimization. Given the practical challenges of this discrete combinatorial optimization, the authors propose an approximate solution using continuous relaxation.

The experimental results indicate that SAE’s continuous relaxation-based sampling yields a uniform similarity distribution in the test set, providing a more accurate estimate of model performance in testing. Additionally, the paper demonstrates that when the distribution of an external test set is known, the optimization can be adapted to align with this specific distribution instead of a uniform one.

**Strengths:**

- The paper is highly readable, with well-organized arguments, making it a thought-provoking study not only for the specific task of DTA but also for general evaluation considerations in molecular machine learning.
- Instead of focusing on prediction methods, this paper addresses the critical issue of how machine learning predictions should be evaluated appropriately. For tasks like drug-target binding affinity (DTA) prediction, practical utility is the key metric, and the gap between benchmarking scores and real-world applicability has been a long-debated issue. It is valuable that such research is presented at venues like ICLR.
- In standard molecular task benchmarking, prediction performance is often evaluated using random split (or scaffold split, as also compared in the paper) for dividing training/validation/test sets. The authors support with data that if an evaluation independent of similarity to the training set is desired, current evaluation practices of using random split or scaffold split may yield an overly optimistic assessment.
- Specifically for the DTA task, the authors empirically demonstrate that regardless of the similarity metric, dataset, prediction target (e.g., IC50 or Ki), or molecular feature representation, as long as random split is used, (1) molecules similar to those in the training set will dominate the validation set, and (2) consequently, only the prediction accuracy for training-set-similar molecules affects the overall performance score. The prediction accuracy for low-similarity molecules, which are fewer in number, is often overlooked. However, considering the actual needs in DTA, this point cannot be ignored.
- As an alternative to the random split, the authors propose a similarity-aware evaluation (SAE), defining it as a combinatorial optimization problem that considers similarity bins and aims to create a uniform distribution within each bin. They also propose a practical solution for SAE using continuous relaxation to approximate the discrete combinatorial optimization.

**Weaknesses:**

- The technical novelty of the work is relatively limited, as the proposed SAE is essentially a specific case of "stratified sampling" from statistics. The "similarity bins" that underpin SAE effectively serve as strata defined by similarity, meaning that if the dataset is pre-divided into strata based on molecule similarity, applying random sampling within each stratum would naturally achieve uniformity across similarity bins. Therefore, it seems feasible to implement this approach more directly without relying on the combinatorial optimization and continuous relaxation proposed by the authors. From this perspective, the combinatorial optimization formulation may appear to overly complicate the problem.
- In molecular machine learning, including general QSAR tasks, the challenge of fair predictive evaluation has been a longstanding issue. For instance, it was highlighted as a concern for inadequate QSAR model validation over a decade ago [1]. Additionally, this challenge is closely related to out-of-distribution (OOD) generalization [2], and thus it is not exclusive to the DTA task. In [2], a similar analysis to this paper is presented, introducing a "deployment-to-train" distribution-based splitting protocol that resembles the approach proposed in this paper. Note that [2] does not propose any new splitting method, which does not diminish the contribution of this paper.
  - [1] Cherkasov A, Muratov EN, Fourches D, Varnek A, Baskin II, Cronin M, Dearden J, Gramatica P, Martin YC, Todeschini R, Consonni V, Kuz'min VE, Cramer R, Benigni R, Yang C, Rathman J, Terfloth L, Gasteiger J, Richard A, Tropsha A. QSAR modeling: where have you been? Where are you going to? J Med Chem. 2014 Jun 26;57(12):4977-5010.
  - [2] Tossou P, Wognum C, Craig M, Mary H, Noutahi E. Real-World Molecular Out-Of-Distribution: Specification and Investigation. J Chem Inf Model. 2024 Feb 12;64(3):697-711. doi: 10.1021/acs.jcim.3c01774.

**Questions:**

Have you tried a direct approach like this (stratified sampling), where the dataset is first divided into K bins based on similarity to other molecules (stratification), and then random sampling (random splitting) is applied within each bin to create a test set that is uniform at  similarity-based bins?

---

> ### Author Response · Authors · 2024-11-23
>
> Thank you for your comments. We really appreciate your emphasis of the importance and challenge of fair predictive evaluation in molecular machine learning. In fact, it is also our concern that the majority of literature is seen to use the randomized split to construct the testing set based on which the possibly overoptimistic performance of their proposed models is measured and reported. This is just what motivates this work from the very beginning. Thank you for pointing out the relevant works and we have added discussions on these works to the Section A.1 of the revised version.
>
> With respect to your suggestion on the approach of stratified sampling, in the stratification stage each molecule needs to be calculated a similarity metric based on which strata are divided. However, the similarity is pairwise and an aggregation is thus needed across all other molecules, for example by averaging or taking the maximum. There may be a problem that the aggregation is taken over all other molecules, since what matters most are only molecules that are not in the same subset. Consider an extreme example of some molecule in the testset, suppose it has many high-similarity counterparts but they are accidentally all in the testset, while molecules in the trainset are all less similar to this molecule. In this case the molecule should be regarded as low-similarity to the trainset but the aggregation of similarities across all other molecules may be high. Stratified sampling will consequently fail in such cases, while SAE will not, since it carefully differentiate, for each molecule, molecules that are in the same subset from those are not, and aggregates across only molecules that are not in the same subset, as in Eq.(5) and (6). We regard this as the major difference between SAE and stratified sampling. We also perform experiments showing that they have indeed performance gap, as illustrated in Table A.1 of the appendix of the revised version.
>
> > Comparison against other existing split strategies, detailed results are shown in Table A.1 of the appendix
> | Split | Method | External Test MAE | External Test R2 |
> | - | - | - | - |
> |Random| FusionDTA | 1.1198 | 0.0957
> || ChemBERTa | 1.0600 | 0.1517
> || MolCLR | 1.0976 | 0.1546
> || PharmHGT | 1.1107 | 0.1323
> || SAM-DTA | 0.9863 | 0.3206
> |Scaffold| FusionDTA | 1.0863 | 0.1243
> || ChemBERTa | 1.1972 | -0.0491
> || MolCLR | 1.1585 | 0.0405
> || PharmHGT | 1.0930 | 0.1594
> || SAM-DTA | 1.0187 | 0.3034
> |SIMPD| FusionDTA | 1.1417 | 0.0528
> || ChemBERTa | 1.1010 | 0.0878
> || MolCLR | 1.2131 | -0.0016
> || PharmHGT | 1.1588 | 0.0133
> || SAM-DTA | 1.0058 | 0.3083
> |Stratified (max)| FusionDTA | 1.0886 | 0.1517
> || ChemBERTa | 1.1504 | 0.0223
> || MolCLR | 1.0917 | 0.1207
> || PharmHGT | 1.0694 | 0.1811
> || SAM-DTA | 1.0345 | 0.2722
> |Stratified (avg)| FusionDTA | 1.0957 | 0.1206
> || ChemBERTa | 1.1258 | 0.0896
> || MolCLR | 1.1556 | 0.1019
> || PharmHGT | 1.0938 | 0.1667
> || SAM-DTA | 0.9946 | 0.3345
> |SAE (mimic)| FusionDTA | 1.0605 | 0.2122
> || ChemBERTa | 1.0452 | 0.2477
> || MolCLR | 1.0002 | 0.2981
> || PharmHGT | 1.0609 | 0.1861
> || SAM-DTA | 0.9773 | 0.3367

---

> > ### Comment · Reviewer_PwZH · 2024-11-25
> >
> > Thank you for the clarification! You’re absolutely right that with simple stratified sampling, the issue of how to aggregate similarities arises. I now understand this point well. Since you also reported the accuracy validation using max and avg, I’ve gained a much clearer understanding of the value of the proposed method.

---

### Official Review · Reviewer_dEEq · 2024-11-02

**Soundness:** 3
**Presentation:** 3
**Contribution:** 2
**Rating:** 6
**Confidence:** 4

**Summary:**

This paper addresses a critical limitation of evaluation schema in current drug-target affinity (DTA) prediction models. Traditional methods often employ a randomized dataset split, leading to test sets dominated by samples similar to the training set. This approach causes inflated model performance on high-similarity samples while overlooking the degradation in performance on low-similarity samples, which are common in real-world scenarios. To address this, the authors propose a novel "Similarity Aware Evaluation" (SAE) framework, which introduces an optimized splitting methodology to better match target distributions across similarity bins. This paper demonstrates that SAE enables a more accurate assessment of model generalizability, thereby enhancing performance on external test sets and improving robustness in real-world applications.

**Strengths:**

- The paper introduces the Similarity Aware Evaluation (SAE) framework, an innovative methodology for creating test sets that better represent real-world data distributions.
- The authors conduct extensive experiments across multiple datasets and state-of-the-art models, demonstrating the widespread impact of similarity bias.

**Weaknesses:**

- The continuous relaxation and hyperparameter tuning required in the SAE framework might introduce additional computational complexity and practical challenges for large datasets or high-throughput applications. Could you provide further clarification on the time complexity of the SAE framework, especially in terms of how these processes impact scalability and efficiency in such contexts?
- The performance of the proposed SAE framework likely varies depending on the similarity measurement used, such as Tanimoto or cosine similarity, and may also be influenced by the choice of fingerprint (e.g., ECFP, MACCS, rdikit). A discussion comparing the performance of the SAE framework across these different similarity measures and fingerprints would provide a more comprehensive understanding of its effectiveness.
- While the paper compares SAE with randomized and scaffold splits, it could benefit from a more detailed analysis against other existing methods tailored for similarity-based evaluations, such as time-based, stratified, and cold-drug splits. To provide a clearer benchmark, it would be helpful to include a comparison table showing the performance of SAE against these methods on a common dataset. Additionally, comparisons with specific methods [1,2,3] could be included, or an explanation could be provided for why certain comparisons may not be applicable. This would allow for a more comprehensive evaluation of SAE's effectiveness relative to existing approaches.

[1] Pahikkala, Tapio, et al. "Toward more realistic drug–target interaction predictions." Briefings in bioinformatics 16.2 (2015): 325-337.

[2] Nguyen, Tri Minh, Thin Nguyen, and Truyen Tran. "Mitigating cold-start problems in drug-target affinity prediction with interaction knowledge transferring." Briefings in Bioinformatics 23.4 (2022): bbac269.

[3] Atas Guvenilir, Heval, and Tunca Doğan. "How to approach machine learning-based prediction of drug/compound–target interactions." Journal of Cheminformatics 15.1 (2023): 16.

**Questions:**

Please see the weaknesses.

---

> ### Author Response · Authors · 2024-11-23
>
> Thank you for your comments. With respect to the time complexity, theoretically the computational complexity of SAE is O(M*(N^2)) given the dataset size N and the number of iterations M. Empirically, for the EGFR dataset in the paper which contains 4,361 ligands, with the number of iterations set to 20,000, SAE takes approximately 270 seconds on a single NVIDIA GTX 3090 GPU, with a memory usage of 2,410 MiB. We would like to emphasize that SAE is used in the model development stage and as a result, when the model is developed, it will no longer affect the efficiency for high-throughput inference. Note also that SAE needs only to be performed once for a fixed dataset, meaning that it can be reused by different models as long as they are developed on the same dataset. Therefore, the time it takes will be overwhelmed by the time used by the heavy model development. When scaling to large datasets, it should be noted that almost all operations within one iteration is parallelable, and as a result it will benefit significantly from more powerful GPU devices and distributed computation. We appreciate that you point out the missing analysis of time complexity. We agree that time complexity is of vital importance to practitioners. These discussions have been added in the Section A.3 of the revised version of the paper.
>
> As for comparison across different similarity measures and fingerprints, we conduct experiments on similarity measures including Tanimoto, cosine, Sokal and Dice, and fingerprints including Morgan (ECFP), RDKFP (RDKit) and Avalon. As shown by Table A.3 in the appendix, split results are less affected by similarity measure choices but more influenced by fingerprint choices. In all settings of similarity measures and fingerprints, SAE outperforms other counterparts by achieving a split that is closer to the desired distribution (uniform distribution in this case).
>
> > Comparison across different similarity measures and fingerprints, the desired distribution is a uniform distribution across the bins [0, 1/3], (1/3, 2/3], and (2/3, 1]. The numbers in this table are presented in the form [Sample counts in the first bin, Sample counts in the second bin, Sample counts in the third bin]. Detailed results are shown in Table A.3 of the appendix.
> | Similarity | Fingerprint | SAE (balanced) | Random | Scaffold | SIMPD | Stratified (max) | Stratified (avg)
> | - | - | - | - | - | - | - | -
> |Sokal | Morgan | 292, 289, 292 | 33, 398, 442 | 172, 510, 191 | 689, 126, 58  | 34, 423, 416 | 29, 416, 428
> || RDKFP  | 291, 291, 291 | 19, 80, 774  | 85, 236, 552  | 135, 624, 114 | 14, 82, 777  | 15, 74, 784
> || Avalon | 291, 291, 291 | 7, 63, 803   | 26, 275, 572  | 23, 629, 221  | 8, 76, 789   | 6, 73, 794
> |Tanimoto | Morgan | 290, 299, 284 | 16, 98, 759  | 80, 273, 520  | 228, 547, 98  | 12, 99, 762  | 13, 99, 761
> || RDKFP  | 289, 292, 292 | 8, 34, 831   | 15, 184, 674  | 2, 635, 236   | 1, 39, 833   | 4, 33, 836
> || Avalon | 220, 325, 328 | 0, 28, 845   | 2, 154, 717   | 1, 324, 548   | 0, 30, 843   | 1, 28, 844
>
> As for comparison against other existing approaches, we compare with stratified sampling as shown in Table A.1 of the appendix. SAE performs better than stratified sampling with a clear margin. Note that time-based and cold-drug splits are not applicable, since the time-based split requires temporal information which our datasets do not contain, and the cold-drug split requires multiple proteins, whereas in our experiments each dataset is about a single protein. References [1-3] are also not applicable. Specifically, [1] discusses factors that affect the evaluation results, among which the fourth factor “experimental setting” is the closest to the topic of this paper (whether trainset and testset share common drugs and proteins, only drugs or proteins or neither). However, also because each dataset is about a single protein in this paper, trainset and testset must not share any drugs in all cases. [2] proposes a modeling method for the cold-start problem and thus focuses on model development instead of evaluation. [3] studies many aspects in the drug-target interaction prediction problem, including a comparison between the randomized split and the split that trainset and testset are completely disallowed to share similar drugs. However, this split leads to a testset that is all in the low-similarity bin, incapable of achieving any other distributions such as the uniform distribution among different similarity bins. Performance on mid- and high-similarity bins is thus not able to be measured.

---

> > ### Comment · Reviewer_dEEq · 2024-11-25
> >
> > Thank you for your detailed responses and for addressing the comments provided. I appreciate the effort you have put into revising the manuscript and clarifying the points raised. I have carefully reviewed your responses and the revised manuscript, and I would like to further discuss a few points and share additional suggestions.
> >
> > - I understand that the cold-drug split can be applied to a single protein, and I expect that [3] would yield similar results. As the authors mentioned, I also agree that performance comparisons may not be feasible in the mid-/high-bin scenarios. However, reflecting on this, wouldn’t it be possible to interpret that a model performing well in such a setting is likely to perform effectively on out-of-distribution data? I would like to hear the authors' additional thoughts on this point.
> > - In Figure A.1, most molecules fall into the similarity bin of 1/3 or below, which forms the basis for the discussion in Appendix A.4 on applying the SAE framework to large-scale datasets. However, in Table A.3 or Table 2, it seems that most molecules belong to the bin above 2/3 (apologies if I have misunderstood this). In such cases, is it still possible to apply the SAE framework to large-scale datasets?

---

> ### Author Response · Authors · 2024-11-26
>
> Thank you for your additional suggestions and discussions.
>
> - In our experiments, the cold drug split is degenerated to the randomized split when applied to a single protein, as each sample is characterized by only one type of attribute: the SMILES representation. With respect to [3] that leads to a testset that is all in the low-similarity bin, we agree that it is a good indicator for out-of-distribution data. That is, a model performing well in this setting is likely to perform effectively on out-of-distribution data. However, if on the contrary models perform not so well, which is more probable since low-similarity data are generally more difficult, it would be hard to distinguish a model that is well trained but struggles to solve the low-similarity data from one that is not well trained at all. In Table A.1 of the appendix we added the results of [3] as the dissimilar split. Due to the all-low-similarity split, performance is bad even in the internal testset, indicating that the model selection fails. In such cases, we would argue that a combination of low-, mid- and high-similarity data in the testset is better for selecting the model.
>
> - We are sorry but there may be a misunderstanding. In Table 2 and Table A.3, counts in the bins are based on the **maximum** similarity across molecules in the trainset, for each molecule in the testset (please also refer to Eq.(5-6)); whereas in Figure A.1, the histogram is calculated over all entries in the pairwise similarity matrix, without the maximum operation. This is the reason why they may be different.

---

> > ### Comment · Reviewer_dEEq · 2024-11-26
> >
> > Thank you for the additional discussions and clarifications. I am happy to increase the my score.

---

### Official Review · Reviewer_C9Wr · 2024-11-03

**Soundness:** 3
**Presentation:** 2
**Contribution:** 3
**Rating:** 8
**Confidence:** 4

**Summary:**

The paper focus on analysing the evaluation methods on Drug-Target binding Affinity (DTA) prediction. The paper first shows that state-of-the-art methods for drug target affinity prediction tend to perform better on drugs that are similar to the drugs used to train the models, than on those with a lower similarity. Authors also shows that current sampling strategies used for the evaluation of DTA tend to report a performance value that is much closer to the similar drugs in training, while the performance for dissimilar drugs is much worst.

Authors then propose a split methodology called Similarity Aware Evaluation (SAE). The proposed split strategy is capable of reflect any desired distribution, by formulating the problem as an optimisation problem, that can be approximated by gradient decent.

**Strengths:**

The paper presents thorough evaluations on current SOTA methods for DTA, showing that performance is quite skewed by similar drugs used for training. It clearly illustrates the difference in performance at different levels of drug similarity, under different representations and with several similarity measures.

The proposed similarity aware evaluation and the math behind it is well explained and overall easy to understand.

The experiments to validate the proposed evaluation are reasonable. The influence and impact of the parameters controlling it are reasonably justified.

**Weaknesses:**

The paper needs to be proof read for English. While the main ideas in the mathematical part can be understood, the paper writing needs to be improved. (too many missing articles, for example)

While the results presented by the authors do indeed show that current DTA prediction methods tend to perform better on drugs with high similarity comparing to the ones used for training and poorly on the ones with low similarity, this fact is hardly novel or surprising. The influence and challenges of similarity in drug related tasks has been previously documented, but not referenced in the paper. In QSAR related literature this is a known fact, and other areas predicting quantitative properties of drugs do control for this. For example:

Sheridan, Robert P., et al. "Similarity to molecules in the training set is a good discriminator for prediction accuracy in QSAR." Journal of chemical information and computer sciences 44.6 (2004): 1912-1928.

In other areas, such as Drug-Target Interaction prediction this is also often reported by measuring performance on dissimilar drugs in the testing set, for example:

Luo, Yunan, et al. "A network integration approach for drug-target interaction prediction and computational drug repositioning from heterogeneous information." Nature communications 8.1 (2017): 573.

Wan, Fangping, et al. "NeoDTI: neural integration of neighbor information from a heterogeneous network for discovering new drug–target interactions." Bioinformatics 35.1 (2019): 104-111.

This is similar to what authors report on Figure 1 on the left column. All being said, authors do a good job quantifying and reporting the difference in performance with respect to the similarity.

The related works section itself is reduced and quite compressed, but most of the related work is covered in the introduction.

In the abstract and in the text, the use of the word “target” in the sentence “target distribution” is confusing in the context of this paper, since it could be easily confused with “target protein distribution”.

**Questions:**

What exactly do authors mean when writing "samples are allowed to coexist in trainset and testset"? (lines 84 and 311) One possible interpretation of this sentence could be that a sample can be used for training the model and later also used to validate the performance. This seems not to be the right interpretation given several parts of the mathematical section of the paper, yet this sentence made me wonder if this is indeed the case.

In terms of reporting performance, what are the advantages of using SAE over reporting the performance on the different similarity brackets (as done in Figure 1, left column)? This would allow to better pick the model that better suits the similarity bracket instead of having one model for all the drugs.

Related to the previous question, it would be interesting to see the performance on the different brackets in the external dataset of figure 5. Does using SAE improves the performance across all the similarity brackets?

The research topic could also be extended to other QSAR tasks, and while not needed, a small commentary about this could enrich section "4.3 Other Applications".

Do authors consider providing the code implementing SAE, and the code to replicate the experiments?

---

> ### Author Response · Authors · 2024-11-23
>
> Thank you for your comments. We fully agree with you on the importance of quantifying and reporting the difference in performance with respect to the similarity. This may have been investigated in some works for drug related tasks, as you have mentioned, but unfortunately the majority of ongoing literature is still seen to use the randomized split to construct the testing set based on which the performance of their proposed models is measured and reported. We believe that part of the reason is the lack of a practical alternative split strategy that can adapt to any desired distribution. Reporting the performance on the different similarity brackets is surely a better way, but it also suffers from the fact that low-similarity brackets contain too few samples under the randomized split, meaning that the performance measured may have large variance and thus is less trustable, as illustrated by Figure 1. We therefore argue that a completely redesigned split strategy is still needed and this is just the original motivation of this work.
>
> We appreciate that you point out the missing references. We have already added them and elaborated the related work Section A.1 in the appendix of the revised version. We have followed your suggestions and added detailed results of the performance on the different brackets in the external dataset, in the appendix Table A.2. As is shown, SAE improves performance in the low- and mid-similarity brackets, but not in the high-similarity one. We think this is because internal testset by SAE puts more samples in the low- and mid-similarity brackets and thus performance in these brackets receives more attention during model development, compared with other split strategies. We have also added commentary about extension to other QSAR tasks in Section 4.3 and Section A.2 of the revised version. The paper is proof read the paper and possibly confusing word uses such as“target” are refined. The code will surly be released upon publication, to facilitate the reproducibility and future research.
>
> > MAE score in the external test set, detailed results can be found in the appendix Table A.2.
> | Bin       | Count | Split            | FusionDTA | ChemBERTa | MolCLR  | PharmHGT | SAM-DTA |
> |-----------|-------|------------------|-----------|-----------|---------|----------|---------|
> | [0, 1/3]  | 120   | Random           | 1.2442    | 1.1167    | 1.2385  | 1.3261   | 1.1343  |
> |           |       | Scaffold         | 1.2293    | 1.2242    | 1.1630  | 1.1840   | 1.1363  |
> |           |       | SIMPD            | 1.4297    | 1.3223    | 1.7376  | 1.3382   | 1.1179  |
> |           |       | Stratified (max) | 1.2311    | 1.1663    | 1.2414  | 1.2179   | 1.1747  |
> |           |       | Stratified (avg) | 1.5371    | 1.4478    | 1.6005  | 1.3962   | 1.3666  |
> |           |       | SAE (mimic)      | 1.0626    | 1.0315    | 1.1848  | 1.1082   | 1.0435  |
> | (1/3, 2/3]| 1026  | Random           | 1.1416    | 1.0874    | 1.0983  | 1.0989   | 0.9879  |
> |           |       | Scaffold         | 1.0729    | 1.2273    | 1.2021  | 1.0891   | 1.0209  |
> |           |       | SIMPD            | 1.1545    | 1.1272    | 1.2134  | 1.1614   | 1.0158  |
> |           |       | Stratified (max) | 1.0756    | 1.1611    | 1.0927  | 1.0677   | 1.0317  |
> |           |       | Stratified (avg) | 1.0815    | 1.1450    | 1.1384  | 1.0835   | 0.9808  |
> |           |       | SAE (mimic)      | 1.0531    | 1.0594    | 0.9979  | 1.0606   | 0.9743  |
> | (2/3, 1]  | 186   | Random           | 0.9559    | 0.9060    | 1.0208  | 1.0516   | 0.9015  |
> |           |       | Scaffold         | 1.0604    | 1.0346    | 0.9449  | 1.0543   | 0.9335  |
> |           |       | SIMPD            | 0.9741    | 0.9022    | 1.0066  | 1.0775   | 0.9191  |
> |           |       | Stratified (max) | 1.0718    | 1.0883    | 1.0171  | 0.9933   | 0.9683  |
> |           |       | Stratified (avg) | 0.9694    | 0.9047    | 1.0415  | 1.0102   | 0.8967  |
> |           |       | SAE (mimic)      | 1.1003    | 0.9762    | 0.8938  | 1.0322   | 0.9511  |
>
> With respect to the sentence "samples are allowed to coexist in trainset and testset" (lines 84 and 311),  we are explaining the continuous relaxation used to solve the combination optimization problem. Note that Eq.(6) in the formulation of the combination optimization problem constrains that the trainset and the testset are disjoint. However, the constraint is discrete and makes the problem infeasible to solve for optimum. To solve it approximately, we propose the continuous relaxation where each sample is assigned a weight w_i which denotes the “proportion” the sample belongs to the testset. This is the reason why we say that samples are allowed to coexist in trainset and testset. Note that after solving the problem, we sort the samples by the weights and take only the top \alpha N samples as the testset. As a result, the final split is guaranteed not to have any sample that coexists in trainset and testset.

---

> > ### Comment · Reviewer_C9Wr · 2024-11-27
> >
> > I thank the authors for their thorough responses to my questions. I will raise my score accordingly.
> >
> > However, there are still a couple of grammatical mistakes that should be corrected for the final version. I will include a couple of them as examples:
> >
> > In the whole document, testset and trainset are written as a single word. This is correct in many fields, but is often found in academic papers as “test set” and “training set”. This is a suggestion to improve formality and readability, authors could also keep ‘testset’ and ‘trainset’ in the text, but it should be noted that this is not the standard practice.
> >
> > Across the whole text, authors mix passive and active voice. This results in a paper that is often tedious to read and confusing.
> >
> > In line 92, it says "we have managed to achieve testset split with various", it should be either "we have managed to achieve a testset split with various" or "we have managed to achieve testset splits with various..." depending on what authors intend to say.
> >
> > in line 140-141 it says "is general for randomized split of testset across different similarity measures" and should be " is general for a randomized split of a testset across different similarity measures" or something similar. In the same line, often "first" is preferred over 'firstly', but 'firstly' is also fine, up to authors.
> >
> > in line 212 it should be " because the majority of samples in the trainset"
> >
> > then, in the same sentence, it should be “the results are shown in Table 1. Here’
> >
> > in line 270 it should be. “The other is PharmHGT” and not “Another”
> >
> > In line 274, ‘at’ is used instead of  ‘in’
> >
> > In line 275 it should say ‘targets’ instead of ‘target’.
> >
> > In line 278 “The results are collectively presented in Table 2. Here”
> >
> > In line 279, “The” should be lowercase.
> >
> > In line 288, “which aims at testset with desired distribution”,  a verb is missing, “aims at finding a’ or ‘obtaining a’ and then ‘with a desired distribution’
> >
> > In line 291 “The split for a testset” instead of “The split for testset”

---

> > > ### Author Response · Authors · 2024-11-28
> > >
> > > Thank you and we really appreciate that you kindly point out these grammatical mistakes. We are sorry for the confusion and inconvenience caused by our inappropriate word uses and grammatical mistakes. We have corrected all that you have mentioned, and carefully proof read and refine the whole text as we can, in our updated version. Thank you!

---

### Meta-Review · Area_Chair_MoV8 · 2024-12-17

**Metareview:**

This paper proposes a similarity-aware evaluation (SAE) approach for splitting datasets in drug-target affinity prediction, aiming to produce test sets with controlled similarity distributions. While not drastically new conceptually (in my opinion), it does demonstrate practical utility by showing that standard random splits can yield misleading performance estimates when low-similarity samples are rare -- which is important!

The authors did respond in detail to reviewer comments and made efforts to clarify certain points, address missing references, and promise code release. They also added new results in appendices as requested. Appreciate for their effort.

By the end of the discussion, the reviewers generally aligned on a positive stance. There were no major disagreements left unresolved. Two common points raised by multiple reviewers were: (1) The need to consider different similarity measures and fingerprints. After revision, the authors showed that SAE consistently outperforms other methods across various similarity metrics and fingerprints. (2) Concerns about complexity and scalability. The authors addressed this by providing approximate runtimes and memory usage, and also potential ways to handle large datasets (e.g., sparse matrices).

Given that the authors addressed the key criticisms (data complexity, comparisons with various methods, and improved clarity), and the reviewers ended up supportive, acceptance seems justified.

**Additional Comments On Reviewer Discussion:**

The authors did respond in detail to reviewer comments and made efforts to clarify certain points, address missing references, and promise code release. They also added new results in appendices as requested.

There were no major disagreements left unresolved. Two common points raised by multiple reviewers were: (1) The need to consider different similarity measures and fingerprints. After revision, the authors showed that SAE consistently outperforms other methods across various similarity metrics and fingerprints. (2) Concerns about complexity and scalability. The authors addressed this by providing approximate runtimes and memory usage, and also potential ways to handle large datasets (e.g., sparse matrices).

---

### Decision · Program_Chairs · 2025-01-22

Accept (Oral)